# AMMI analysis of elite bread wheat (*Triticum aestivum* L.) selections for genotype by environment interaction and stability of grain yield in Southern Ethiopia

**Liyew Alemayehu**[ID][�], **Mesfin Kebede**[ID]*[�], **Eyasu Wada**[ID]

Wolaita Sodo University, Sodo, South Ethiopian Region, Ethiopia

[�u̯] These authors contributed equally to this work.
* mesfin04@yahoo.com

**Data Availability Statement:** All relevant data are within the manuscript and its Supporting Information files.

## Abstract

Smallholder wheat farmers of Ethiopia frequently use landraces as seed sources that are low yielders and susceptible to diseases due to shortage of seeds of adapted improved bread wheat varieties. Developing novel improved varieties with wider adaptability and stability is necessary to maximize the productivity of bread wheat. Hence, a multi-location field trial was conducted across four locations in south Ethiopia during the 2022/23 main cropping season with the objective of estimating the magnitude of genotype by environment interaction (GEI) effect, and determine the stable genotype among the 10 Ethiopian bread wheat advanced selections using a randomized complete block design (RCBD) with three replications. The data recorded from all plots on 13 agronomic traits and the three wheat rust diseases were computed using appropriate statistical software. The results showed that individual and combined analysis of variance (ANOVA) exhibited the presence of highly significant variability (P<0.01) among the locations, genotypes, and GEI effects for most of the traits including grain yield. The additive main effects and multiplicative interaction (AMMI) ANOVA for main effects; location, genotype and GEI revealed significant variation among the selections with 82.0%, 8.7% and 9.3% share of sum square variation, respectively. The genotype plus genotype by environment interaction (GGE) bi-plot analysis explained 92.44% of the total variation observed. AMMI and GGE-biplot analyses indicated G11, G9, G10, and G8 as high yielders and well-adaptive in the favourable locations. AMMI stability value (ASV) and Yield stability index (YSI) showed G5 and G8 as highly stable and adaptive selections across locations. Overall, the study identified that G8 as the most stable and adaptive selection, while G11 was the top yielder cultivar across locations. Therefor it was suggested that seeds of G8 can be grown across all the locations, whereas G11, G9, and G10 can be grown in the favourable environments and similar agro-ecologies in the east African region.

**Funding:** The author(s) received no specific funding for this work.

## 1. Introduction

Common wheat (*Triticum aestivum* L.), also called bread wheat is a self-pollinating annual plant in the *Poaceae* family [1]. More than 90% of the world's wheat grain is obtained from bread wheat, a hexaploid species, with three genomes AABBDD [2]. The remaining amount of wheat produce (<10%) is contributed by tetraploid (AABB) durum wheat (*Triticum durum* or *Triticum turgidum ssp. durum*) and emmer wheat (*Triticum dicoccum* L.) [3]. Bread wheat is recognized for its wider adaptability to diverse environments, due to its rich genetic diversity [4]. This adaptability has made it one of the most widely cultivated cereal crops globally, serving as a significant source of dietary energy and nutrition for human consumption [5].

Bread wheat grain is utilized in various forms such as bread, cakes, biscuits, bakery products, and numerous confectionery items. Additionally, its straw is used for animal feed and in paper manufacturing industries [6]. In most developing nations, wheat bread serves as the primary food and source of dietary energy and protein [7]. In the East African region, bread wheat accounts for 9% of daily protein intake and 10% of daily calorie intake. However, due to low domestic yields and rising demand, most African countries (including Ethiopia) import significant quantity of the wheat required for domestic wheat supply, highlighting the need for strong and stable agriculture [8].

Before the 1990s, durum wheat dominated the agricultural landscape in Ethiopia, covering 65–70% of the total cultivated area of the wheat crop. The remaining 30–35% was occupied by bread wheat. However, recent studies have indicated a shift in wheat cultivation patterns in Ethiopia, with bread wheat now surpassing durum wheat in terms of cultivated area [9]. This transformation is primarily attributed to the introduction and widespread adoption of new bread wheat varieties known for their disease resistance, enhanced grain yield and wider adaptability [10].

Bread wheat is grown primarily in the highlands of Ethiopia [11]. It is produced largely in the southeast, central, and northwest parts of the country [12]. Oromia region produces about 59% of the country's wheat, while Amhara region contributes around 27%, the southern regions accounted for 9% and the remaining 5% comes from the Tigray region [13,14]. Another study indicated that about 46% of the 13 million hectares classified as highly suitable for wheat production in the country is found in Arsi and Shewa sub- regions [15].

The success of plant breeding programs depends on making sure farmers have access to seeds of improved varieties that consistently perform well in terms of yield and quality across diverse environmental conditions [16]. Matching genotypes with environments to enhance desirable production is a key objective in plant breeding, since fluctuations in environments directly affect genotypic and additive values [17]. While some genotypes perform well under a variety of conditions, others excel only under specific environmental conditions. This idea of genotype-specific adaptation is closely related to the phenomenon of genotype-environment interaction (GEI). GEI exists when the relative phenotypic performance of genotypes varies with changing environmental factors [18]. A measure of the quantitative traits of plants involves evaluating phenotypic value influenced by genotype, environment, and GEI. Multi-environment trials (METs), evaluating different genotypes in different environments (year, location, or a combination), primarily aim to recommend genotypes suitable for specific environments or delimiting huge environments [19].

Several statistical methods have been developed for METs to study genotype, environment, and GEI effects [20]. Analysis of existing GEI patterns using linear-bilinear models such as additive main effects and multiplicative interactions (AMMI), genotype plus genotype-environment (GGE) models, is essential for defining target environments for future testing and selection of genotypes [21]. AMMI model is useful in understanding the GEI and is broadly

used to analyze stability and adaptability in grain yield [22], by using principal component axis (PCA) scores and AMMI stability values (ASV) across environments [23] to enhance the reliability of the MET analysis [24]. It has also been used to determine the stability of wheat varieties across environments using principal component axis (IPCA) scores [25].

The application of AMMI stability value (ASV) and yield stability index (YSI), as stability measurements is essential because they are proven to have strong associations with cultivar superiority measure (Pi), Wricks ecovalence (Wi), yield stability index (YSI), and regression coefficient (βi) [26]. AMMI analysis clarifies the GEI effects by summarizing the nature and associations of genotype and environments [27]. In addition, it increases the exactness of yield estimates corresponding to increasing the number of replicates by a factor of two to five that may be used to reduce costs by decreasing the number of replications and /or test environments, to include more treatments in the experiment, or to improve efficiency in selecting the best genotypes [28].

The AMMI model combines the analysis of variance for the genotype and environment main effects with principal component analysis of the GXE interaction. It has proven useful for understanding complex GXE interactions. The results can be graphed in a useful bi-plot that shows both main and interaction effects for both genotypes and environments. The bi-plot main effects of means vs. the first interaction principal component analysis axis (IPCA1) from AMMI analysis was used to study the patterns of response of G, E and GEI. It was also used to identify genotypes with broad and specific adaptation to target environments for grain yield [29].

The potential of AMMI analysis in small grains like bread wheat have been demonstrated by [30] who suggested that the GEI effect of the 20 bread wheat genotypes with six environments was best estimated by the first two interaction principal components (IPCA1 and IPCA2) of genotypes and environments. Further, they suggested that bi-plots generated using genotypic and environmental scores of IPCA1 and IPCA2 can be used by breeders and have an overall picture of the behavior of the genotype, the environment and GEIs. Another finding reported by [31] demonstrated that bread wheat genotypes evaluated in multi-environment yield trials by using GGE bi-plot analysis determined the presence of two proper rain fed mega environments.

Recent reports about productivity of wheat in Ethiopia showed that the average grain yield of wheat at the national level was 2.67 t ha$^{-1}$, and 2.51 ton ha$^{-1}$ in the south Ethiopia region [32]. Therefore, there is a pressing need to systematically evaluate the stability and performance of bread wheat varieties across different environments in South Ethiopia. This study seeks to address this knowledge gap by investigating the stability of bread wheat advanced selections in terms of grain yield performance across diverse environmental conditions, aiming to identify selections with superior adaptability and consistent performance. The findings of this research will inform wheat breeding programs, agricultural extension services, and policy-makers, to make use of the best performing selections to be considered as candidates for the registration, release and dissemination of wheat varieties that are resilient and productive across the region and other locations with similar agro-ecological conditions.

## 2. Materials and methods

### 2.1. Descriptions of the study area

A multi-environmental field trial using 10 bread wheat advanced selections and an improved cultivar 'Alidoro' was conducted to evaluate their performance for yield and yield-related traits on smallholder farmer's fields across four districts (Humbo, Sodo Zuria, Boloso Sore, and

Damot Gale) in the South Ethiopia region during the 2022/23 crop season under rain-fed conditions.

Sodo Zuria Woreda is located in the center of Wolaita zone encircling the town of Sodo. This Woreda (district) is located at an altitude ranging between 1500 meters at Zala Shasha kebele and 2,950 meters above sea level at the tip of mount Damot. It covered an area of 46,006 hectare and has 31 kebeles with total population of 178,890 persons. Most of the Woreda's land coverage is midland (Woynadega) except the highland areas of mount Damot that experience colder climate. Wheat grows mainly around mount Damot where sub-tropical climatic conditions dominate. The livelihood of the population is mainly agriculture, where '*enset*' *Ensete ventricosum* L. is the dominant staple food crop. The other important crops grown are root crops such as sweet potatoes, taro, Irish potato and carrot etc., cereals such as maize, wheat, barley, teff, sorghum, and pulses like faba broad bean and field pea grown surrounding mount Damot.

The administrative center of Humbo is Tebela. The agro–ecology of the study *Woreda* consists of 70 percent lowland or *Kola* (below 1500 m.a.s.l.) and 30 percent intermediate highland or *Woina Dega* (between 1600–1900 m.a.s.l.). The major crops cultivated in the Woreda are cereals (teff, maize, and sorghum), pulses (haricot bean and chickpea), and root crops (potato, sweet potato and enset).

Areka is the town of Boloso Sore where the administrative center is located. The Woreda was classified as 83 percent of the land *weyna dega or mid-land* and 17% Dega/highland (above 2000 m.a.s.l.). Major crops grown in Boloso Sore includes root crops; enset, taro, sweet potato, Irish potato; cereals such as, teff, maize, sorghum, and pulse crops,; haricot bean, field pea, cowpea, etc., tree crops like Arabica coffee, vegetables and fruits like banana, mango, papaya are commonly grown. Based on topographic and climatic characteristics, Damot Gale woreda is divided into two major agro-ecological zones; highland (*Dega*) and lowland (*Kola*) constituting 26%, and 74% of the land, respectively. Agriculture dominates 90% of the economic activity in the study area. The major crops grown in the study area include cereals such as teff (*Eragrostis tef* (Zucc.) Trotter), maize (*Zea mays* L.), bread wheat (*Triticum aestivum* L.), haricot bean (*Phaseolus vulgaris* L.), field pea (*Pisum sativum* L.), potato (*Solanum tuberosum*), sweet potato (*Ipomea Batatas*), taro (*Colocasia esculenta*), enset (*Ensete ventricosum*) and coffee (*Coffea arabica*).

Details of the specific information of the four field trial sites are described in Table 1. Data on geographic coordinates and elevation were obtained from GPS measurements, while the temperature and rainfall data were obtained from Ethiopian Meteorological Institute/National Meteorological Agency, Government of Ethiopia (https://www.ethiomet.gov.et/). The soil physic-chemical properties of the four districts (Woredas) are depicted in Table 2.

**Table 1. A specific description of the test locations in the study area.**

| Study location | | Geographic coordinates | | Average annual temperature (˚C) | | Average annual rainfall (mm) | | Altitudes in m.a.s.l |
|---|---|---|---|---|---|---|---|---|
| Districts | Kebele | Latitude | Longitude | Min | Max | Min | Max | |
| Sodo Zuria | Kokate | 6˚52'40"N | 37˚48'30"E | 13.5 | 23.0 | 639 | 1252 | 2163 |
| Damot Gale | Woshgale | 6˚54'39"N | 37˚49'1"E | *11.5* | *21.0* | *1200* | *1300* | 2238 |
| Boloso sore | Gulumo-koysha | 6$^0$05'0"N | 37$^0$0'0"E | 14.5 | 28.5 | 600 | 1330 | 2100 |
| Humbo | Ampo-kysha | 6$^0$44'10"N | 37$^0$48'25"E | 15.0 | 29.0 | 700 | 1000 | 1762 |

**Table 2. Soil properties of the four districts of Wolaita zone that affect growth and development of agricultural crops.**

| Soil properties | Sodo Zuria | Damot Gale | Boloso Sore | Humbo |
|---|---|---|---|---|
| Soil texture | Clay loam | Clay | Loam | Sandy loam |
| Bulk density | 1.33 g cm$^{-3}$ | 1.16 g cm$^{-3}$ | 1.39 g cm$^{-3}$ | 1.08 g cm$^{-3}$ |
| Total porosity | 51.51% | 57.5% | 49.41% | 58.0 |
| Moisture content | 18.14% | 18.05% | 18.52% | - |
| Water holding capacity | 21.43% | 22.06% | 22.78% | - |
| pH | 5.3 | 5.8 | 6.0 | 6.94 |
| Total Nitrogen (%) | 0.117663 | 0.100854 | 0.15 | 0.06 |
| Organic carbon (%) | 1.365 | 1.17 | 1.30 | 0.64 |
| Available phosphorous (ppm) | 12.2 | 9.83 | 7.44 | 3.0 |
| Available Sulfur (ppm) | 16.3 | 15.2 | 18.25 | - |
| Available Boron (ppm) | 0.79 | 0.61 | 0.68 | - |
| CEC(cmol/kg) | 22.4 | 17.5 | 17.6 | 20.0 |

## 2.2. Experimental materials and design

The 10 advanced bread wheat selections (Table 3) were the latest outcome of a wheat variety development research project that started with evaluation of 48 bread wheat landrace accessions sourced from the Ethiopian Biodiversity Institute (EBI) and screened for disease resistance and grain yield by the researchers of Wolaita Sodo University, department of plant science with the collaboration of Debre-Zeit and Kulumsa agricultural research centers' staff and their facilities. Following evaluation of the results only 24 lines were selected for the next cycle of progeny evaluation by the same team of researchers. Subsequently the 24 advanced lines were subjected for preliminary yield trial (PYT) in Wolaita zone at two locations for one year. Based on their performances for grain yield and resistance to wheat rust diseases from the PYT, 10 advanced lines and a standard check 'Alidoro' were chosen for this multi-environmental varietal trial (MEVT). Performances of the selected advanced bread wheat lines and the standard check in the PYT is presented in Table 4.

The MEVT testing sites were selected in the smallholders' farmers' field at four different districts in the South Ethiopian region to test their stability for grain yield and test their adaptability to the respective environments. The land was cleared, plowed twice and leveled before

**Table 3. Descriptions of the bread wheat advanced selections used in the present study.**

| S. N | Bread wheat genotypes | Initial code | Original seed source | Source of seeds of the advanced selections |
|---|---|---|---|---|
| 1 | WSU-BW1 | 226930 (G1) | EBI | WSU |
| 2 | WSU-BW2 | 227248 (G2) | EBI | WSU |
| 3 | WSU-BW3 | 238133 (G3) | EBI | WSU |
| 4 | WSU-BW4 | 238139 (G4) | EBI | WSU |
| 5 | WSU-BW5 | 238498 (G5) | EBI | WSU |
| 6 | WSU-BW6 | 238507 (G6) | EBI | WSU |
| 7 | WSU-BW7 | 238525 (G7) | EBI | WSU |
| 8 | WSU-BW8 | 238543 (G8) | EBI | WSU |
| 9 | WSU-BW9 | 238871 (G9) | EBI | WSU |
| 10 | WSU-BW10 | 238873 (G10) | EBI | WSU |
| 11 | Alidoro | Alidoro (G11) | HARC | HARC |

EBI–Ethiopian Biodiversity Institute; HARC–Holetta Agricultural Research Center; WSU-Wolaita Sodo University; BW- Bread wheat

**Table 4. Mean performance of the 11 bread wheat genotypes grown at Ade Koysha and Kokate in 2021 crop season.**

| SerNo. | Genotype (Code) | DH | DM | BM | TSW | GY | HI | PH | SL | TNT | ENT | KPS | SPS | YR | LR | SR |
|---|---|---|---|---|---|---|---|---|---|---|---|---|---|---|---|---|
| | | | | | | DAMOTE-GALE (ADEKOYSHA site) | | | | | | | | | | |
| 1 | 226930 (G1) | 68 | 114 | 13.73 | 35.3 | 3.94 | 29.5 | 131.4 | 9.3 | 9.5 | 9.0 | 43.9 | 17.7 | 2.5 | 2.2 | 2.2 |
| 2 | 227248 (G2) | 62 | 117 | 10.77 | 27.9 | 2.89 | 27.2 | 88.3 | 8.5 | 8.1 | 7.7 | 35.5 | 16.3 | 3.0 | 2.7 | 2.2 |
| 3 | 238133 (G3) | 64 | 111 | 10.32 | 28.3 | 2.89 | 26.1 | 111.1 | 9.4 | 10.3 | 8.8 | 34.2 | 16.5 | 3.5 | 2.7 | 3.5 |
| 4 | 238139 (G4) | 67 | 117 | 10.0 | 28.5 | 2.77 | 27.7 | 112 | 8.7 | 11.0 | 9.2 | 38.5 | 15.6 | 3.5 | 2.0 | 2.2 |
| 5 | 238498 (G5) | 65 | 104 | 8.5 | 25.1 | 1.62 | 19.0 | 99.2 | 9.9 | 11.1 | 10.3 | 26.5 | 20.2 | 4.0 | 3.0 | 2.0 |
| 6 | 238507 (G6) | 63 | 103 | 7.73 | 21.8 | 1.69 | 24.0 | 105.2 | 10.4 | 10.9 | 9.6 | 36.5 | 20.9 | 3.7 | 4.0 | 2.0 |
| 7 | 238525 (G7) | 64 | 106 | 8.41 | 22.5 | 1.45 | 17.1 | 93.8 | 9.3 | 9.2 | 8.5 | 34.2 | 19.6 | 3.2 | 4.0 | 2.0 |
| 8 | 238543 (G8) | 63 | 101.5 | 7.45 | 19.7 | 1.83 | 22.7 | 99.2 | 10.0 | 9.9 | 8.5 | 32.0 | 18.3 | 5.5 | 4.0 | 2.0 |
| 9 | 238871 (G9) | 59.5 | 100 | 17.73 | 28.2 | 2.98 | 17.3 | 102.9 | 10.1 | 9.4 | 8.0 | 33.4 | 18.8 | 3.5 | 3.5 | 3.0 |
| 10 | 238873 (G10) | 59.5 | 100 | 11.36 | 28.2 | 2.53 | 23.5 | 108.4 | 10.6 | 10.1 | 9.5 | 33.2 | 19.4 | 4.0 | 3.5 | 2.5 |
| 11 | Alidoro (G11) | 66 | 117 | 13.64 | 33.8 | 4.0 | 29.3 | 86.0 | 12.0 | 8.5 | 7.4 | 63 | 21.8 | 1.5 | 3.5 | 1.5 |
| | | | | | | SODO-ZURIA (KOKATE Site) | | | | | | | | | | |
| 1 | 226930 (G1) | 70 | 120 | 11.59 | 38.5 | 1.08 | 9.3 | 127.1 | 12.6 | 9.6 | 9.2 | 44.3 | 19.5 | 2.0 | 2.7 | 2.2 |
| 2 | 227248 (G2) | 65 | 116 | 9.00 | 29.5 | 1.51 | 16.7 | 85.5 | 8.9 | 9.5 | 8.9 | 38.6 | 17.8 | 3.5 | 3.2 | 2.7 |
| 3 | 238133 (G3) | 67 | 118 | 10.91 | 28.4 | 2.63 | 22.8 | 110.5 | 10.5 | 9.4 | 8.7 | 41.7 | 17.4 | 2.5 | 4.2 | 4.2 |
| 4 | 238139 (G4) | 71 | 121 | 12.73 | 31.8 | 3.06 | 24.0 | 117.1 | 10.8 | 8.7 | 8.2 | 49.7 | 17.5 | 2.7 | 3.2 | 2.7 |
| 5 | 238498 (G5) | 72 | 122 | 11.36 | 32.8 | 2.72 | 24.0 | 112.7 | 9.3 | 8.3 | 8.2 | 40.7 | 19.5 | 2.2 | 4.2 | 2.7 |
| 6 | 238507 (G6) | 71 | 121 | 12.50 | 33.4 | 2.61 | 20.7 | 112.6 | 10.15 | 8.3 | 7.8 | 43 | 20.9 | 2.0 | 3.7 | 3.0 |
| 7 | 238525 (G7) | 72 | 123.5 | 10.45 | 32.6 | 2.65 | 25.3 | 111.2 | 10.6 | 8.7 | 8.0 | 43.7 | 20.8 | 2.0 | 4.0 | 2.5 |
| 8 | 238543 (G8) | 71 | 122.5 | 10.00 | 37.1 | 2.27 | 22.9 | 109.7 | 11.5 | 8.7 | 7.9 | 37.4 | 20.3 | 2.5 | 3.5 | 2.7 |
| 9 | 238871 (G9) | 67 | 117 | 9.55 | 37.6 | 2.50 | 26.5 | 109.4 | 11.4 | 8.1 | 7.7 | 39.2 | 19.8 | 1.5 | 3.7 | 3.0 |
| 10 | 238873 (G10) | 67[f] | 116 | 7.73 | 37.6 | 1.96 | 25.6 | 102.7 | 10.5 | 7.4 | 6.9 | 37.2 | 18.4 | 2.0 | 3.7 | 2.5 |
| 11 | Alidoro (G11) | 70 | 116 | 8.64 | 40.3 | 2.36 | 27.3 | 93.9 | 12.7 | 8.6 | 8.5 | 69.2 | 22.3 | 2.5 | 2.5 | 1.7 |

DH = Days to 50% Heading, DM = Days to 90% maturity, BM = Above ground Biomass, TSW = Thousand Seed Weight, GY = Grain yield per hectare, HI = Harvest index, PH = Plant height, SL = Spike length, TNT = Total number of tillers per plant, ENT = Number of effective tillers per plant, KPS = Number of kernels (grain) per spike, SPS = Number of spikelets per spike, YR = Yellow rust, LR = Leaf rust and SR = Stem rust.

sowing. The field trial layout was arranged in a randomized complete block design (RCBD) with three replications at each of the four locations. Each genotype was assigned randomly to the experimental plots within the blocks divided into 11 plots. Each plot consisted of six rows with 2.5 m length and 1.2 m width. There was 0.5 m space between each plot; 20 cm inter-row spacing and the 3 blocks (replications) were separated by 1 m distance. Each experimental plot covered 3 m$^2$ (1.2 m × 2.5 m) and the total experimental areas for each location were 172.9 m$^2$ (9.5 m x 18.2 m).

## 2.3. Experimental field management

Seeds were sown manually by drilling method at a rate of 150 kg ha$^{-1}$ and covered lightly with soil. The full dose (150 kg ha$^{-1}$) of blended inorganic fertilizer NPSB (18.9% N, 37.7% P$_2$O$_5$, 6.9% S, 0.1% B) applied at planting while urea (46% N) was applied in split application (half at sowing time and the remaining at tillering-stage) at a rate of 100 kg ha$^{-1}$ [33]. No chemical and/or cultural control measures were used to control weeds, insect pests and fungal diseases. Weeds were controlled manually as needed.

**2.3.1 Data collection.** All necessary data were collected on plant and plot bases from the four middle rows of each plot excluding the border rows from both sides following the

protocols described in [32,33]. For the data collected on plant basis; viz., number of total tillers per plant (NTTP), effective number of tillers (ENT), number of kernels per spike (NKS), number of spikelets per spike (NSS), plant height (cm)(PH), and spike length (cm)(SL); ten plants per plot were randomly selected and tagged for data recording. For those agronomic parameters, i.e., plant phenological and yield parameters, the number of days to 75% heading (count) (DH), number of days to 90% maturity (count)(DM), grain filling period (count)(GFP), thousand-kernel weight (gm)(TKW), aboveground biomass (t ha$^{-1}$)(AGBM), grain yield (t ha$^{-1}$) (GY), and harvest index (%)(HI); data were recorded on plot basis from all the plants found in the central four rows. Data on severity of the 3 wheat rusts; leaf rust (LR), yellow rust (YR), and stem rust (SR) diseases were also collected on a plot basis every week when disease severity reached between sixty and one hundred percent on susceptible plants in the farmers' fields that grow local or susceptible landrace varieties using a 1 to 9 scoring scale (Table 5) described in [34]. The numerical and symbolic conversions are; 1 = VR (very resistant) with 5% severity; 2 = R (resistant) with10% severity, 3 = R–MR (resistant to moderately resistant) with 20% severity, 4 = MR (moderately resistant) with 30% severity, 5 = MR–MS (moderately resistant to moderately susceptible) with 40% severity, 6 = MS (moderately susceptible) with 60% severity, 7 = MS–S (moderately susceptible to susceptible) with 70% severity, 8 = S (susceptible) with 80% severity, and 9 = VS (very susceptible) with over 90% severity.

Table 5. Rust response assessment scale: A 1–9 rating system for evaluating rust reactions in field conditions.

| Scoring scale 1–9 | Description of rust response | Yellow rust | Leaf rust | Stem rust |
|---|---|---|---|---|
| 1 | Very resistant | No disease | No disease | No disease |
| 2 | Resistant | <5% necrotic dots | <5% necrotic dots with/without occasional medium sized pustules | <5% small sized ruptured uredinia on plant parts |
| 3 | Resistant to moderately resistant | 10–15% necrotic stripes, scattered and rarely sporulating, top leaves often free from infection | 10–15% infected area, pustule size and response may vary between genotypes | 10–15% medium size susceptible pustules often ruptured. Rupturing more pronounced near the node |
| 4 | Moderately resistant | 20–30% necrotic to light sporulating stripe | 20–30% infected area, pustule size and response may vary between genotypes | 20–30% small to medium pustules. Rupturing of some pustules may occur |
| 5 | Moderately resistant to moderately susceptible | 35–45% blotchy island with light sporulation, can have moderate to high chlorosis and necrosis, sporulating island often appear all over the leaf but in some causes restricted toward the tip | 35–45%% infected area, pustule size and response may vary between genotypes with light to medium sporulation | 35–45% medium to large sized moderately sporulating pustules, again more pronounced near the node |
| 6 | Moderately susceptible | 50–60% large stripes often sporulating, stripes can turn necrotic under windy and warmer conditions. Stripe in this category remain distinct and largely uncoalesced | 50–60% susceptible pustules, pustules size may vary | 50–60% susceptible pustules with larger pustules near the node and relatively smaller pustules on the rachis |
| 7 | Moderately susceptible to susceptible | 65–75% sporulating stripe starts to coalesce, chlorosis present | 65–75% susceptible pustule with moderate sporulation | 65–75% susceptible pustule and coalescing of pustules occur, small holes on stem may be visible |
| 8 | Susceptible | 80–90% moderate to heavy sporulation, there is always small green island left near the base of the leaf | 80–90% susceptible pustule with heavy sporulation | 80–90% moderate to heavy sporulation. Medium to large holes on stem visible near nodes and can cause breakage of stems |
| 9 | Very susceptible | 95–100% infection and heavy sporulation leading to defoliation | 95–100% heavy sporulation leading to defoliation | 95–100% heavy sporulation leading to defoliation breakage of stem do to large hole |

Source: [34].

### 2.4. Data analysis

**2.4.1 Analysis of variance.** The data were statistically analyzed using GenStat software (version 18.1, 64-bit) for analysis of variance. The ANOVA was computed for all the agronomic and disease traits considered in the study following the standard steps stated in [35] to assess the performance of bread wheat genotypes for each trait and location. The error variance was calculated for each location. The homogeneity of error variances was tested using Bartlett's test before computing the combined analysis of variance. Majority of the traits except harvest index, leaf rust, and yellow rust, showed homogeneity of variances across the tested locations. The data of traits that showed non-homogenous error variance across locations were transformed by the logit equation [36].

The ANOVA model used for individual sites was

$$Y_{ij} = \mu + G_i + B_j + E_{ij} \tag{1}$$

Where: $Y_{ij}$ = observed value of i$^{th}$ genotype in j$^{th}$ replication, $\mu$ = grand mean of the experiment, $G_i$ = effect of genotype i, $B_j$ = replication effect B$_j$, j = 1...b, (b = number of replications), $E_{ij}$ = error effect of genotype i in replication j.

The ANOVA model used to analyze the data for combined locations was

$$Y_{ijk} = \mu + G_i + L_j + R_{k(j)} + GL_{ij} + E_{ijk} \tag{2}$$

Where: $Y_{ijk}$ = observed value of genotype 'I' in replication k of location 'j', 'μ' = grand mean, $G_i$ = effect of genotype i, $L_j$ = effect of location j, $R_{k(j)}$ = effect of replication k in location j, $GL_{ij}$ = the interaction effect of genotype i with location j, $E_{ijk}$ = error (residual) effect of genotype i in replication k of location j.

For the traits that showed significant differences among the genotypes (at the individual location) and genotype by environment interaction (over location), mean separation test was computed by using Duncan's multiple range tests (DMRT) at the 5% level of significance.

**2.4.2 Additive main effects and multiplicative interaction (AMMI) analysis.** This analysis was performed using the statistical software GenStat (VSN International Ltd) according to the formula proposed by [37]:

$$Y_{ij} = \mu + G_i + E_j + \sum_1^N \lambda_k \, Y_{ik} \, \delta_{jk} + \varepsilon_{ij} \tag{1}$$

Where: $Y_{ij}$ is the grain yield of the i$^{th}$ genotype in the j$^{th}$ location, μ is the grand mean, $G_i$ and $E_j$ are the genotype and location deviation from the grand mean, respectively, λk is the eigenvalue of the PCA axis k, Υik and δjk are the genotype and location principal component scores for axis k, N is the number of principal components kept in the model and Ɛij is the residual term. The IPCA axis degrees of freedom (df) are calculated using the following simple method of [27].

$$DF = G + E - 1 - 2n; \tag{2}$$

Where: G = the number of genotypes, E = the number of locations, and n = the n$^{th}$ axis of IPCA.

**2.4.3 AMMI stability value (ASV) and Yield stability index (YSI).** AMMI stability value (ASV) and yield stability index (YSI) were performed using the statistical software package agricolea [38] of R-Studio version 4.3.0 as described by [39] as follows:

$$ASV = \sqrt{\left[\frac{SSIPCA1}{SSIPCA2} * (IPCA1 \; scores)\right]^2 + IPCA2 \; scores^2} \tag{3}$$

Where; ASV = AMMI stability value, IPCA1 = interaction principal component analysis 1, IPCA2 = interaction principal component analysis 2, SSIPCA1 = sum of squares of the

interaction principal component one, and SSIPCA2 = sum of squares of the interaction principal component two.

The grain yield stability of 11 bread wheat genotypes tested at four locations was evaluated using Kang's yield stability index [40]. This analysis, which included both mean yield and stability, was performed using the following formula:

$$YSI = RASV + RY, \tag{4}$$

Where: RASV is the rank of AMMI stability value and RY is the rank of mean grain yield of genotypes across locations.

**2.4.4 Genotype plus genotype by environment interaction (GGE) biplot analysis.** This analysis was performed on the grain yield basis using the statistical software tool GenStat 64-bit version 18.1 according to the method described by [41]:

$$Y_{ij} - \mu = \lambda_1 f_{1i}{}^n{}_{1jk} + \lambda_2 f_{2i}{}^n{}_{2j} + e_{ij} \tag{5}$$

where $Y_{ij}$ is the mean for the $i^{th}$ genotype in the $j^{th}$ location, $\mu$ is the overall mean, $\lambda 1$ and $\lambda 2$ are the singular values of the first and second principal components (PC 1 and PC 2) respectively, $f_{1i}{}^n$ and $f_{2i}{}^n$ are the eigenvectors for the $i^{th}$ genotype for PC 1 and PC 2 respectively, 1j and 2j are the eigenvectors for the $j^{th}$ location for PC 1 and PC 2 respectively and $e_{ij}$ is the residual error term.

# 3. Results

## 3.1. Analysis of variance

Mean squares of the Univariate ANOVA computed using the data recorded from the treatments for the 16 traits including the 3 major wheat diseases, stem, leaf, and yellow rust for each of the 4 locations are presented in the S1–S3 Tables. Majority of the traits; DH, DM, GFP, PH, SL, NKS, NSS, TKW, GY, and HI showed significant differences (P<0.05 or P<0.01) among the bread wheat selections at each of the four individual locations.

Analysis of variance for the combined data across the four locations is presented in Table 6. The outcome of the combined ANOVA indicated that 15 (93.75%) of the traits showed significant ($p<0.05$; $p<0.001$) differences among the selections except for leaf rust disease severity data whereas, the mean squares due to locations/environments demonstrated highly significant ($p<0.001$) differences among the bread wheat selections for all traits. On the other hand the traits such as DH, DM, PH, NSS, NKS, TKW, HI, AGBM, LR, YR, and GY showed significant interaction effects between locations and genotypes (Table 6). The result indicated these traits were influenced by both the bread wheat genotypes and the existing environmental conditions.

## 3.2. Comparison of the average performance of the 10 bread wheat selections and the standard check 'Alidoro' for the 16 traits evaluated across locations

The phenological traits; DH, and DM discriminated among the bread wheat selections that indicated the genotypes could be classified in to 3 categories; viz. early, late and intermediate stages. G4 was identified as both late heading (73.6 days) and late maturity (118.8 days) type; while G11 (61.67) and G2 (62.42) could be considered as early heading types; following a similar trend G9 (107.0) and G10 (109.8) could be categorized as early maturing types (Table 7). Mismatching of days to heading versus days to maturity for most genotypes indicated the variations in their grain filling period.

**Table 6. Mean squares from combined ANOVA across locations for grain yield and yield components including disease parameters with their corresponding CV for the 11 bread wheat genotypes tested across locations.**

| Traits | df | DH | DM | GFP | PH | NTTP | ENT | NSS | NKS | SL | TKW | HI | AGBM | LR | YR | SR | GY |
|---|---|---|---|---|---|---|---|---|---|---|---|---|---|---|---|---|---|
| Replication | 2 | 3.84 | 20.93 | 46.45 | 52.85 | 0.94 | 1.71 | 10.26 | 66.99 | 0.41 | 11.98 | 0.096 | 8.12 | 0.11 | 0.053 | 0.05 | 1.06 |
| Genotypes | 10 | 199.59** | 96.74** | 133.02** | 894.50** | 14.7* | 9.4* | 28.07*** | ** | 8.12** | 135.73** | 0.24** | 8.72** | 0.097$^{ns}$ | 0.85** | 6.57** | 1.67** |
| Location | 3 | 734.15** | 8710.73** | 4828.05** | 18189.22** | 17.553* | 60.93** | 33.38*** | 933.93** | 29.41** | 1975.37** | 1.54** | 433.58** | 1.24** | 0.72** | 2.7$^{ns}$ | 52.16** |
| GxL | 30 | 24.83** | 23.57* | 18.16$^{ns}$ | 91.05** | 3.68$^{ns}$ | 3.84$^{ns}$ | 3.52* | 40.67* | 0.72$^{ns}$ | 24.99** | 0.11** | 3.36* | 0.11* | 0.45** | 1.21$^{ns}$ | 0.59** |
| Error | 86 | 7.68 | 11.03 | 15.56 | 27.81 | 4.92 | 4.37 | 2.14 | 24.30 | 0.73 | 2.99 | 0.02 | 1.61 | 0.06 | 0.07 | 1.36 | 0.18 |
| CV% | | 4.1 | 3.0 | 8.9 | 5.6 | 29.1 | 31.1 | 8.3 | 12.2 | 8.9 | 5.0 | 13.5 | 16.2 | 8.9 | 10.8 | 21.7 | 17.9 |

Total df = 43

Where:—**—Significant at P<0.01

*—Significant at P<0.05, and ns—the non-significant, number in parenthesis represented the degree of freedom for the respective source of variation. DH = days to heading, DM = days to maturity, GFP = grain filling period, PH = plant height (cm), SL = spike length (cm), NKS = number of kernels spike$^{-1}$, NSS = number of spikelets spike$^{-1}$, NTTP = number of total tillers plant$^{-1}$, ENT = effective number of tillers plant$^{-1}$, SR = stem rust(scale), LR = leaf rust, YR = yellow rust (scale), GY = grain yield (t ha$^{-1}$), ABM = biomass yield (t ha$^{-1}$), HI = harvest index, TKW = thousand kernel weight (g).

**Table 7. Mean values of the different agronomic traits for the 11 genotypes across the four locations in 2022/23.**

| Genotypes | DH | DM | GFP | PH | NTTP | ENTP | NSS | NKS | SL | TKW | HI | ABM | LR | YR | SR | GY |
|---|---|---|---|---|---|---|---|---|---|---|---|---|---|---|---|---|
| 226930 (G1) | 65.67[c] | 112.4[bc] | 46.58 | 110.23[a] | 6.68 | 5.77 | 17.25[c] | 42.33[b] | 9.14 | 37.67[ab] | 25.51[e] | 8.78[a] | 6.25 | 10.21[bc] | 5.4 | 2.31[cd] |
| 227248 (G2) | 62.42[de] | 111.8[bc] | 49.33 | 77.51[f] | 6.38 | 5.65 | 15.22[d] | 35.80[c] | 7.99 | 29.38[g] | 26.78[de] | 5.544[c] | 6.25 | 17.71[a] | 5.6 | 1.45[e] |
| 238133 (G3) | 66.42[c] | 113.5[b] | 47.08 | 101.54[b] | 8.38 | 7.55 | 15.97[d] | 36.68[c] | 9.07 | 32.46[f] | 27.9[cd] | 7.97[ab] | 7.50 | 8.75[bcd] | 7.5 | 2.24[cd] |
| 238139 (G4) | 73.58[a] | 118.8[a] | 45.17 | 102.39[b] | 9.07 | 7.60 | 15.71[d] | 40.34[bc] | 8.76 | 29.79[g] | 24.86[e] | 7.86[ab] | 5.83 | 8.33[cd] | 5.2 | 2.14[d] |
| 238498 (G5) | 71.25[ab] | 112.4[bc] | 41.17 | 93.31[cd] | 7.62 | 6.68 | 18.48[bc] | 39.37[bc] | 9.46 | 35.54[cd] | 28.41[cd] | 7.82[ab] | 6.67 | 7.500[d] | 5.2 | 2.4[bcd] |
| 238507 (G6) | 71.33[ab] | 111.8[bc] | 40.50 | 92.47[cd] | 8.12 | 7.35 | 18.65[b] | 39.42[bc] | 9.64 | 34.21[de] | 27.94[cd] | 7.49[b] | 6.04 | 7.50[d] | 5.0 | 2.31[cd] |
| 238525 (G7) | 71.42[ab] | 113.2[b] | 41.83 | 92.81[cd] | 8.12 | 7.217 | 19.02[ab] | 37.95[bc] | 9.88 | 33.12[ef] | 28.11[cd] | 7.58[ab] | 6.04 | 8.33[cd] | 5.0 | 2.3[cd] |
| 238543 (G8) | 70.17[b] | 111.5[bc] | 41.33 | 90.49[d] | 8.33 | 7.20 | 18.00[bc] | 36.60[c] | 10.44 | 36.42[bc] | 29.18[bc] | 8.29[ab] | 7.71 | 7.71[d] | 5.0 | 2.57[abc] |
| 238871 (G9) | 65.42[c] | 107.0[d] | 41.58 | 92.39[cd] | 7.38 | 6.58 | 18.15[bc] | 36.63[c] | 9.89 | 39.00[a] | 30.81[b] | 8.54[ab] | 6.04 | 7.92[d] | 5.0 | 2.75[ab] |
| 238873 (G10) | 64.42[cd] | 109.8[c] | 44.58 | 95.46[c] | 8.5 | 7.40 | 18.30[bc] | 38.40[bc] | 10.33 | 38.38[a] | 30.73[b] | 8.24[ab] | 5.83 | 7.71[d] | 5.0 | 2.63[abc] |
| Alidoro (G11) | 61.67[e] | 111.0[bc] | 49.33 | 86.17[e] | 5.33 | 5.00 | 20.18[a] | 60.77[a] | 10.84 | 37.67[ab] | 34.92[a] | 8.07[ab] | 5.42 | 10.83[b] | 5.0 | 2.825[a] |
| Range | 61.67–73.58 | 107.0–118.8 | 40.50–49.33 | 77.5–110.23 | 5.33–9.07 | 5.000–7.60 | 15.22–20.18 | 35.80–60.77 | 7.99–10.84 | 29.38–39.00 | 24.86–34.92 | 5.544–8.78 | 5.42–7.71 | 7.500–17.71 | 5.0–7.5 | 1.45–2.83 |
| Mean | 67.61 | 112.1 | 44.4 | 94.07 | 7.63 | 6.73 | 17.72 | 40.39 | 9.586 | 34.88 | 28.65 | 7.83 | 6.33 | 9.32 | 5.4 | 2.357 |
| LSD 5% | 2.249 | 2.696 | 3.201 | 4.28 | 1.8 | 1.697 | 1.187 | 4.0 | 0.692 | 1.403 | 1.999 | 1.029 | ns | 1.994 | 0.95 | 0.343 |
| CV% | 4.1 | 3 | 8.9 | 5.6 | 29.1 | 31.1 | 8.3 | 12.2 | 8.9 | 5 | 13.5 | 16.2 | 29.6 | 10.8 | 21.7 | 17.9 |

Means with the same letters are not statistically significant. Where, DH = days to heading, DM = days to maturity, GFP = grain filling period, PH = plant height (cm), SL = spike length (cm), NKS = number of kernels spike$^{-1}$, NSS = spikelets spike$^{-1}$, NTTP = number of total tillers plant$^{-1}$, ENTP = effective number of tillers plant$^{-1}$, SR = stem rust(scale), LR = leaf rust, YR = yellow rust (scale), GY = grain yield (t ha$^{-1}$), ABM = biomass yield (t ha$^{-1}$), HI = harvest index, TKW = thousand kernel weight (g).

Among the growth traits PH exhibited significant variation between the bread wheat selections ranging from 77.51 cm (G2) to 110.23cm (G1). The top three performing selections (G8, G9, and G10) including cultivar 'Alidoro' exhibited intermediate plant height ranging between 86 cm (Alidoro) to 95 cm (G10). Significant variations were also depicted for the traits; NSS and NKS, in which case, G11 (20.18, 60.77) had the highest number of NSS and NKS, while G2 (15.22, 35.80) displayed the lowest performances in both traits, respectively.

Grain yield (GY) and its component traits; TKW, AGBM, and HI depicted significant differences among the bread wheat selections. The average grain yield ranged from 1.45 t ha$^{-1}$ to 2.83 t ha$^{-1}$ with a mean of 2.36 t ha$^{-1}$. G11 had the highest yield (2.83 t ha$^{-1}$) followed by G9 (2.75 t ha$^{-1}$), G10 (2.63 t ha$^{-1}$), and G8 (2.57 t ha$^{-1}$). Statistically, there was no significant difference between G8, G9, G10, and G11 in terms of grain yield performance. G11 has demonstrated excellent performance in mean GY and other yield-related traits (HI, ABM, TKW, SL, NKS, and NSS) across locations. Contrary to G11, selection G2 registered the lowest GY (1.45 t ha$^{-1}$) and its associated traits viz. ABM, TKW, HI, NKS and NSS including the growth traits PH and SL (Table 7).

The severity of leaf rust disease ranged from 5.42% to 7.71% with a mean value of 6.33%. Most of the selections displayed similar resistance levels (statistically at par) to severity of leaf

rust disease; however, there is a statistically significant difference for G4, G10, and G11 with the remaining genotypes. Similarly, the severity of yellow rust disease ranged from 7.50% to 17.71%, 'with a mean value of 9.32%. More than half of the genotypes showed resistance to moderately resistant levels. Among the tested genotypes, G9 (7.92%), G8 (7.71%), G10 (7.71%), G5 (7.50%), and G6 (7.50%) showed similar responses, suggesting a resistant feature. However, G2 (17.71%) exhibited a moderately resistant level to yellow rust disease. No statistical difference depicted among the tested bread wheat selections. The top four high yielders G11, G9, G10, and G8 exhibited resistance response to all three rust diseases (Table 7).

### 3.3. Additive main effects and multiplicative interaction (AMMI) analysis

The AMMI analysis of variance for grain yield showed highly significant differences for the main effects; genotypes, locations, and GEI with 8.7%, 82%, and 9.3% of the variance due to the sum of squares obtained in that order, respectively (Table 8). To further partition the multiplicative variance of the genotype sum of squares caused by GEI, interaction principal component analysis (IPCA) was employed. The result showed that IPCA1 explained 78.04% of the total variation, while IPCA2 accounted for 16.39% of variation. The ordination technique revealed significant differences for IPCA1 but non-significant for IPCA2 (Table 8).

The IPCA1 scores of all the 11 bread wheat genotypes tested in the four locations are presented in Table 9. Based on this data, the genotype G2 (-1.19) had the highest value of IPCA1 score indicating that this genotype contributed more to the GEI, hence, considered as unstable genotype. Such genotypes could adapt to specific environments that depend on their responsiveness to environmental factors. On the other hand, the genotypes G1 (-0.10), G3 (0.06), G4 (-0.20), G5 (0.18), and G8 (0.21) had the lowest IPCA1 score values hence; contributed less to the GEI implying they were stable genotypes (Table 9).

### 3.4. Stability and adaptability analysis

Figs 1 and 2 show bi-plot graphs of the AMMI1 (IPCA1 vs genotypes and environment additive effects) and AMMI2 (IPCA2 vs IPCA1) models, respectively, which help to visualize the relationships between the genotypes and locations. The AMMI 1 biplot graph displayed the superior and highest-yielding genotypes (G5, G8, G9, G10, and G11) on the right side of the graph. In similar fashion, the locations Damot Gale, Boloso Sore, and Kokate were positioned on the right side of the graph indicating they are favorable environments since their grain yields were above the locational mean. Additionally, the locations Damot Gale (0.99) and Boloso Sore (-0.87) had the greatest IPCA1 scores, which significantly influenced GEI, implying these experimental locations were high-yielding locations. However, Damot Gale displayed higher stability than Boloso Sore due to its lower IPCA2 scores (-0.47). Meanwhile, Kokate

**Table 8. AMMI Analysis of variance table for GY data collected from across locations.**

| Source of variation | df | Sum of Squares (SS) | Mean Squares | % Total variation (SS) explained | % G x E explained | % G x E cumulative |
|---|---|---|---|---|---|---|
| Genotype | 10 | 16.66 | 1.67** | 8.70 | | |
| Location | 3 | 156.47 | 52.16** | 82.00 | | |
| Interaction | 30 | 17.58 | 0.59** | 9.30 | | |
| IPC1 | 12 | 13.72 | 1.14** | | 78.04 | 78.04 |
| IPC2 | 10 | 2.88 | 0.29$^{ns}$ | | 16.39 | 94.43 |
| residuals | 8 | 0.98 | 0.12$^{ns}$ | | 5.57 | 100.00 |
| Error | 80 | 11.86 | 0.15 | | | |

**Table 9. Mean grain yield, AMMI Stability Value (ASV), and Yield Stability Index (YSI) for 11 bread wheat genotypes tested across four locations in 2022 crop season.**

| Genotypes | Genotype code | Mean (t ha-1) | RGY | ASV | RASV | YSI | RYSI | IPCA1 | IPCA2 |
|---|---|---|---|---|---|---|---|---|---|
| G226930 | G1 | 2.31 | 6 | 0.53 | 2 | 8 | 2 | -0.10 | -0.24 |
| G227248 | G2 | 1.45 | 11 | 5.65 | 11 | 22 | 11 | -1.19 | 0.24 |
| G238133 | G3 | 2.24 | 9 | 0.53 | 1 | 10 | 6 | 0.06 | 0.45 |
| G238139 | G4 | 2.14 | 10 | 0.97 | 4 | 14 | 9 | -0.20 | 0.08 |
| G238498 | G5 | 2.40 | 5 | 0.89 | 3 | 8 | 1 | 0.18 | -0.22 |
| G238507 | G6 | 2.31 | 7 | 1.96 | 10 | 17 | 10 | 0.41 | 0.04 |
| G238525 | G7 | 2.30 | 8 | 1.35 | 6 | 14 | 8 | 0.28 | 0.17 |
| G238543 | G8 | 2.57 | 4 | 1.03 | 5 | 9 | 3 | 0.21 | -0.28 |
| G238871 | G9 | 2.75 | 2 | 1.84 | 9 | 11 | 7 | 0.37 | 0.47 |
| G238873 | G10 | 2.63 | 3 | 1.51 | 7 | 10 | 5 | 0.31 | -0.21 |
| Alidoro) | G11 | 2.83 | 1 | 1.69 | 8 | 9 | 4 | -0.34 | -0.50 |
| Locations | | | | | | | | | |
| Boloso Sore | | 2.71 | 2 | 4.19 | 3 | 5 | 3 | -0.87 | -0.52 |
| Damot Gale | | 3.61 | 1 | 4.75 | 4 | 5 | 2 | 0.99 | -0.47 |
| Humbo | | 0.61 | 4 | 2.47 | 2 | 6 | 4 | -0.50 | 0.41 |
| Kokate | | 2.49 | 3 | 0.59 | 1 | 4 | 1 | 0.38 | 0.57 |

RGY is the rank of the genotypes given based on the average grain yield data across locations

(0.38) contributed less to the interaction variation due to its lower IPCA1 score, resulting in minimal variance between each genotype's average grain yield and the overall mean yield at this location (Table 9).

## 3.5. AMMI 2 biplot

The information generated from IPCA1 and IPCA2 analysis indicated that G4 showed higher stability due to its proximity to the origin, but exhibited low yield. Further, genotypes G2, G11,

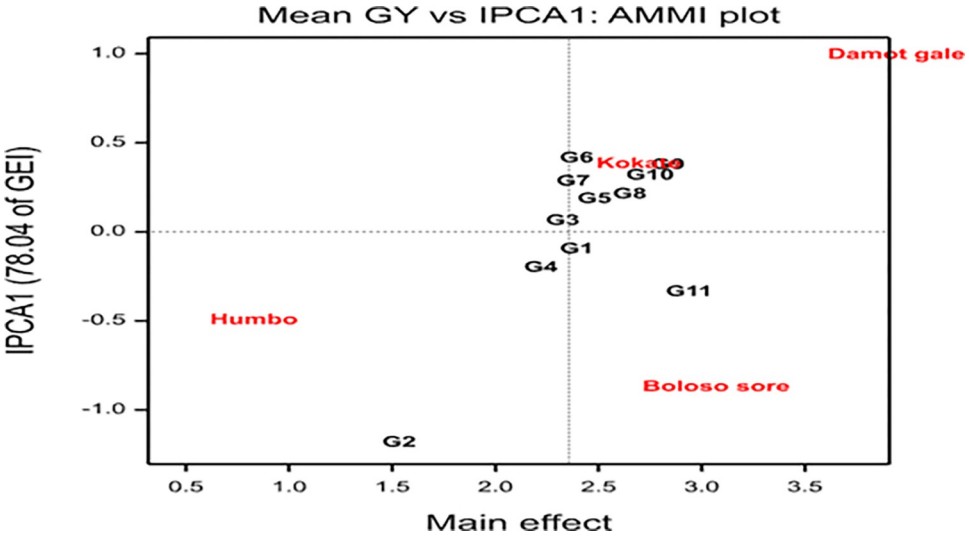

**Fig 1. AMMI 1 Biplot for grain yield (ton ha$^{-1}$) of 11 bread wheat genotypes and 4 locations.**

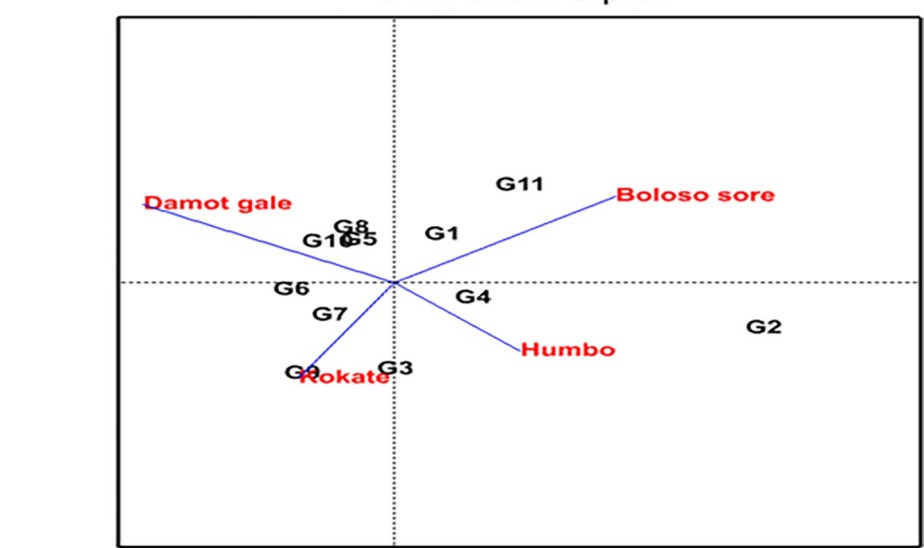

**Fig 2. GEI biplot analysis using the first two interactions' principal component scores from the AMMI 2 model.**

and G9 were unstable (Fig 3), because they were spread out far from the other genotypes and the origin too.

Boloso Sore and Damot Gale stand out as the biggest contributors to the GEI effect, but Humbo and Kokate seemed low. Hence, the average grain yield of each genotype in Kokate and Humbo locations showed minimal variance from the overall mean yield, highlighting their reliability and consistency.

### 3.6. AMMI stability value (ASV)

As presented in Table 8, G2 scored the maximum AMMI Stability Value (ASV) for grain yield, indicating G2 was the most unstable genotype with an ASV value of (5.65). Accordingly, G3 > G1 > G5 > G4 > G8 in that order were the top five most stable and adaptive genotypes based on their respective score of ASV.

### 3.7. Yield stability index (YSI)

Among the genotypes evaluated, G1, G5, G8, G10, and G11 were the top-ranking (best five) genotypes according to Yield Stability Index (YSI), showing both stability and grain yield performance. However, G2 showed the highest instability and low grain yield performance, which was supported by highest ASV (5.65), YSI (22), PC1 (-1.19) values presented in Tables 3 and 4.

### 3.8. Genotype plus genotype by environment interaction (GGE) biplot analysis

The GGE bi-plot analysis explained 92.44% of the total variation observed, of which 76.13% was explained by the first principal component (PC1), while the second principal component (PC2) revealed 16.31% (Fig 3). A graphical analysis is presented for the identification of the winner genotype within the mega environment by visualizing interaction patterns between

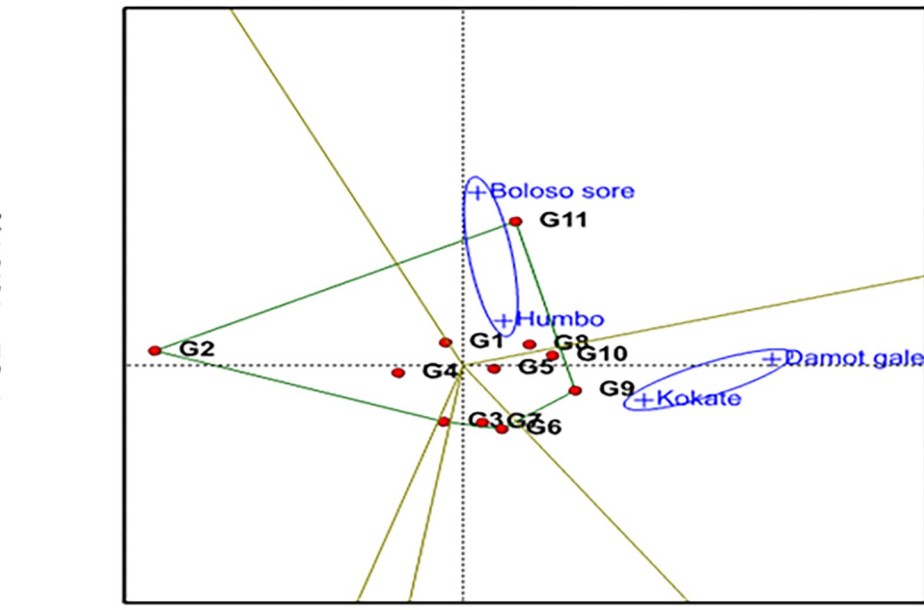

**Fig 3. The GGE biplot approach in polygons for selecting the best genotypes for each mega-environment.**

genotype and environment. In the current work, visualization was done with a genotype-focused scale for genotypic comparison and an environment-focused scaling for environmental comparison (Figs 4 and 5). Additionally, it was decided to visualize the which-won-where pattern of the yield data using symmetric scaling [42].

### 3.9. Which won where biplot view

The outputs obtained from the GGE biplot analysis displayed the genotypes into five-sided polygon vectors based on their relative distance from the biplot origin and divided the graph into five sectors (Fig 3). The locations were distributed across two sectoral areas, but the genotypes were distributed throughout the entire sector.

   The first sector had two locations (Boloso Sore and Humbo, the first mega environment) and two genotypes, G11 and G8. The vertex and best-yielding genotype for this section was G11 (Fig 3). Similarly, two locations (Damot Gale and Kokate, the second mega environment) were found in the second sector. This sector encompassed G9 as the vertex genotype, suggesting that this genotype was the best performer at Kokate and G10 was the top yielder at Damot Gale. Similarly, the third sector had G6 (vertex) and G7, the fourth sector had G3 and the fifth sector had G2 as the vertex genotypes with no specific locations. Thus, genotypes G1, G3, and G5 were close to the origin of the biplot, exhibiting near-average performance, and their GEI variability was smaller than that of the apex genotypes.

### 3.10. Relationship among locations

Using the test for angles between location vectors, three groups of the four locations were formed. It was found that the first group had a minimal angle between "Boloso Sore and

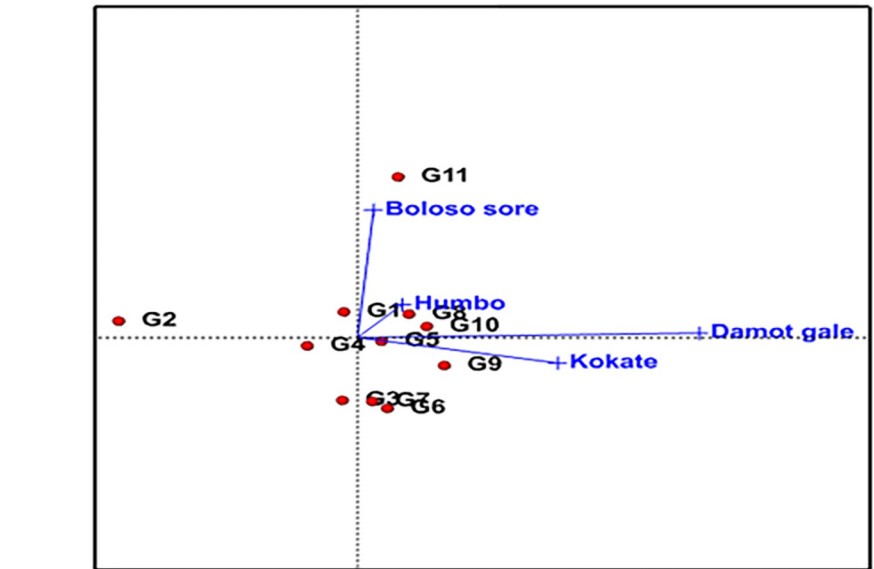

**Fig 4. GGE biplot showing how the four locations are related.**

Humbo", "Humbo and Damot Gale", and "Damote Gale and Kokate", showing they gave similar information about genotypes, and that there was a strong positive association between them (Fig 4). The 2nd group was formed between Boloso Sore and Damote Gale, which is a

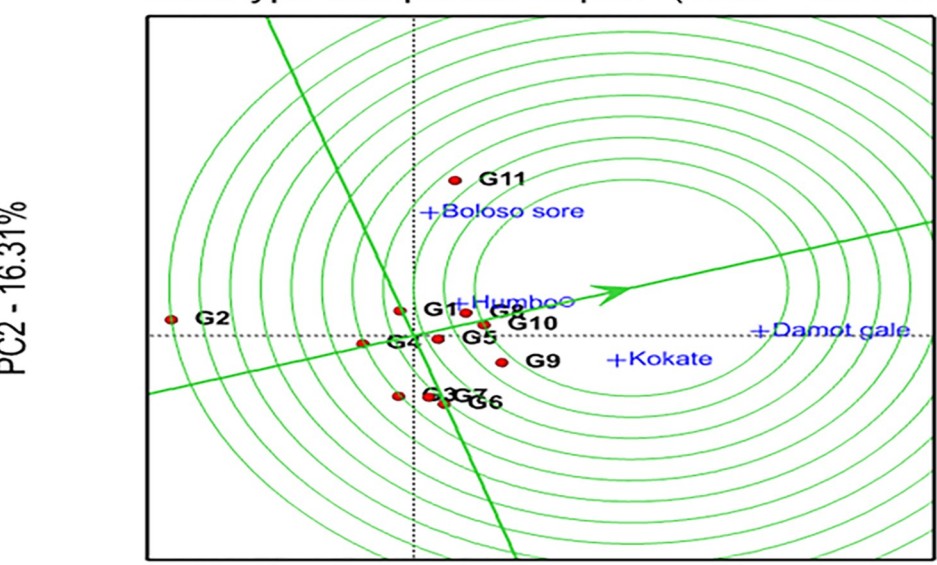

**Fig 5. Ranking of genotypes based on the ideal genotype (in the center of the concentric circles).**

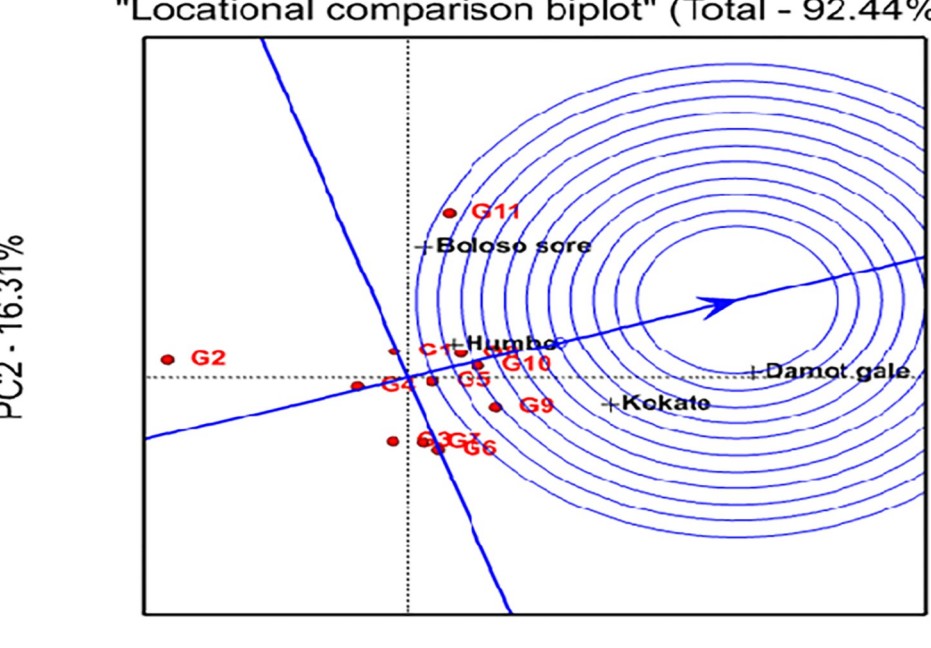

**Fig 6. Positioning of the locations of the trial sites (environment) in order according to the ideal location.**

nearly 90˚ angle between them, while the 3rd group was formed between locations Boloso Sore and Kokate that seems a slightly obtuse angle (>90˚).

### 3.11. Evaluation of genotypes based on the ideal genotype

Analysis of GEIs is important not only for identifying suitable production and testing environments but also for choosing and releasing the best genotypes. In this study the GGE biplot graph (Fig 5) displayed G10 in the first concentric circle which exhibited the highest average grain yield at the ideal location "Damot Gale" (Fig 6), and relatively located closer to it, hence, G10 was considered as the ideal genotype. The genotypes G8, and G9 were relatively closer to the ideal genotype (G10); therefore, they were more preferred over the other tested bread wheat genotypes. The lowest-yielding genotype (G2) was regarded as the undesirable genotype because it was the farthest located from the ideal genotype. Additionally, G1, G3, and G4 were placed near the origin of the biplot, which indicates their resilience to environmental changes.

### 3.12. Evaluation of locations relative to ideal locations

Since the relative position of the environment to the first concentric circle's center determine the ideal testing environment or location, Damot Gale (DG) was regarded as the ideal environment (Fig 6). This environment is located in the 1st concentric circle and has exhibited the highest grain yield (Table 5) displayed the most powerful discriminating ability for the tested bread wheat genotypes. On the other hand, Kokate is the most stable location among all locations, as demonstrated by its significantly low PC1 (0.38), ASV (0.59), and YSI (4) values (Table 9). Hence, DG could be designated as an ideal location and might be a benchmark to rank other locations for bread wheat production (Fig 6).

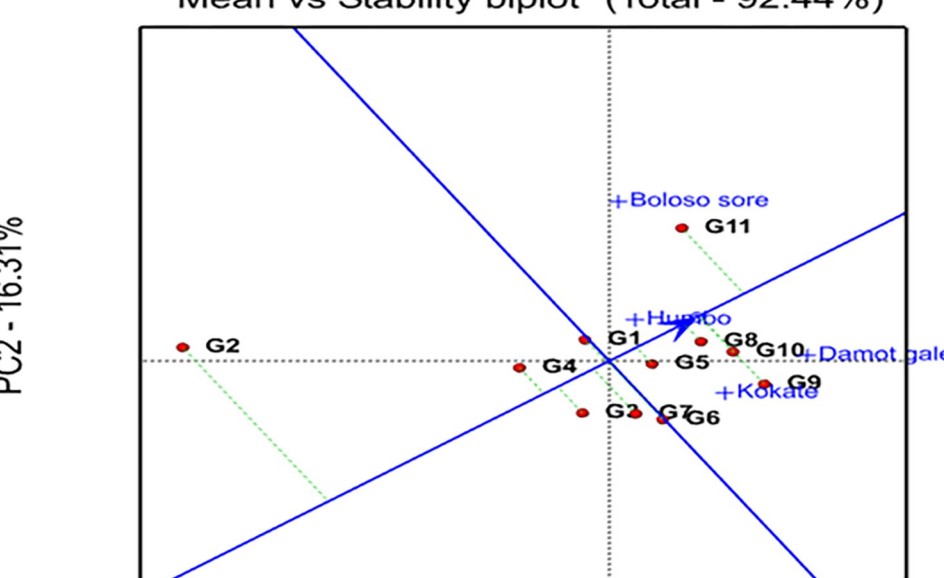

**Fig 7. Distribution of the 11 bread wheat genotypes and the four locations produced by using mean and stability data of GGE biplots analysis of grain yield.**

Kokate is the desired location due to its closeness to the ideal location. In contrast to these locations, Boloso Sore was not a representative location due to its distance from the ideal location required to select specially adapted genotypes for this environment.

### 3.13. Ranking of genotypes based on mean yield and stability performance

The genotype closer to the concentric circle produced more grain on average. The single arrowed line (Fig 7) in the present study indicates high yields in all locations. The genotype with the highest mean yield was G11, followed by G9 > G10 > G8>G5. The average yield for G2 was the lowest and the average yield for G3, G1, G6, and G7 were closest to the overall average. Additionally, genotypes G2 and G11 were unstable genotypes across the locations due to their long perpendicular line from the average environmental coordinate (AEC), while G8 > G5 > G4 > G1 > G3 > G10 were considered relatively stable genotypes in that order; however, they had lower mean grain yield than the check genotype G11. Hence, G8 and G5 were the preferred genotypes due to their stability values and acceptable yield performance.

## 4. Discussion

### 4.1. Analysis of variance

In our investigation of bread wheat advanced selections performance evaluations for grain yield, agronomic parameters and wheat rust disease resistance across locations, we found highly significant variations for the main effects; amongst the selections (G), environments (E), and G x L interactions, for majority of the traits including grain yield (Table 6). Similarly, it was reported that bread wheat genotypes had significant differences in the number of days to maturity, number of days to heading, plant height, spike length, number of kernels per spike, thousand kernel weight, the harvest index, and the grain yield per plant [43].

Several studies conducted in different regions of Ethiopia, including Southern Nations Nationalities Peoples Regional State, Wolaita Sodo, and Dawuro zones, have reported significant variation among bread wheat genotypes for traits such as grain filling period, number of spikelets per spike, number of total tillers/plants, the effective number of tillers/plants, above-ground biomass, yellow rust, and stem rust [32,44], aligning with the findings of the present study.

The analysis of variance using combined data showed non-significant variation for stem rust disease ($p<0.05$) due to locations. In line with the current finding [32] reported a non-significant difference for stem rust due to locations on bread wheat in the Wolaita and Dawuro zones. So, the incidence or severity of stem rust disease is relatively similar across these locations suggesting a potential presence of uniform genetic resistance mechanisms within the tested bread wheat genotypes against prevalent strains of the stem rust pathogen. However, ongoing research and monitoring efforts remain crucial to sustain effective disease management and guide breeding strategies, ensuring long-term resilience against evolving stem rust diseases.

The interaction between genotype and environment plays a significant role in determining the expression and performance of these traits. This suggests that different genotypes may respond differently to varying environmental conditions [45], resulting in variations in the performance of the genotypes to the evaluated traits. In this study the variation among the environments could be depicted in Fig 8 and Table 2, which indicated significant differences have been observed in climatic and edaphic factors, respectively. Hence, we should consider these interaction effects when breeding or selecting varieties for either specific and /or wider environments.

Significant differences in GEI are a symptom that genotypes are inconsistently responding to changing environments because of GEI [46]. Significant GEI can result from changes in the relative importance of genotypes in the environment, or changes in genotype size differences in the environment [47]. But non-significant traits such as stem rust, grain filling period, number of total tillers/plants, the effective number of tillers/plants, and spike length for GEI imply these traits are not affected by the interaction between genotype and environment. This suggests that these traits are relatively stable across the four different growing conditions of the environments evaluated [45].

About 68.75% of traits showed a significant variation for the interaction of genotype by location effect (Table 8), suggesting that these bread wheat genotypes responded differently to the environmental conditions [45], and highlighting the significance of assessing genotypes under various environments to identify better-performing genotypes for wider environmental adaptation. The significant GEI suggests that the agronomic traits of the genotypes varied across the tested environments [47]. Similar to this study, an earlier study conducted on 30 bread wheat genotypes in two heterogenic environments depicted that plant height, number of kernels per spike, above-ground biomass, grain yield, thousand kernel weight, and harvest index showed significant differences in interaction between environments and genotypes [48]. They also reported significant differences in the number of tillers/plants and number of effective tillers/plants, different from the current finding, when subjected to the combined assessment of variance across the three environments.

## 4.2. Comparison of the mean performance of traits among the bread wheat selections evaluated across locations

The study findings indicated there is significant variation in days to heading and days to maturity among the different bread wheat advanced selections in the study area. This variation was

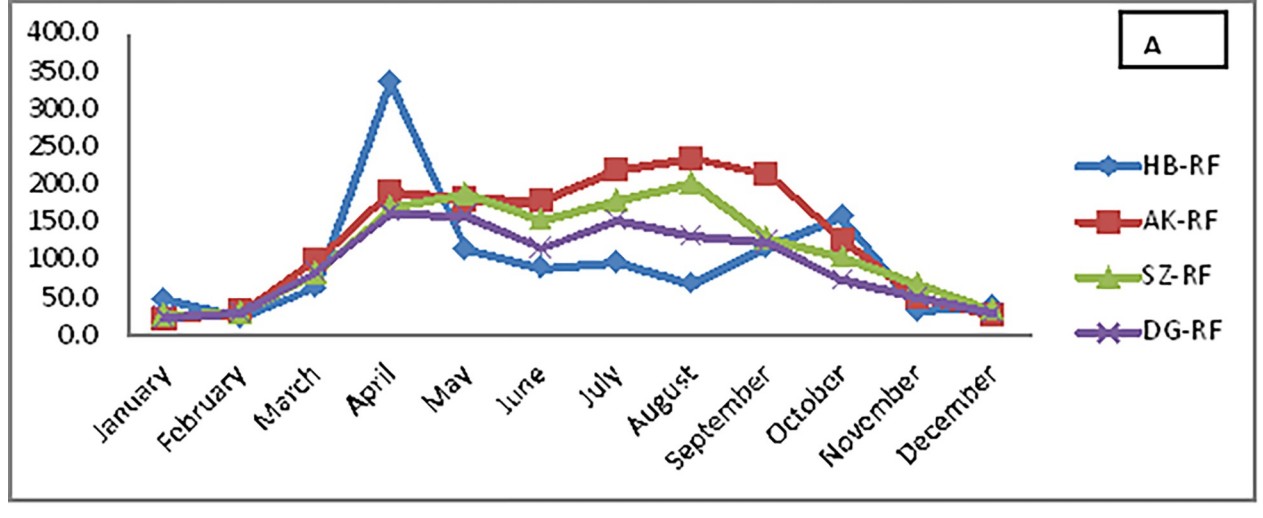

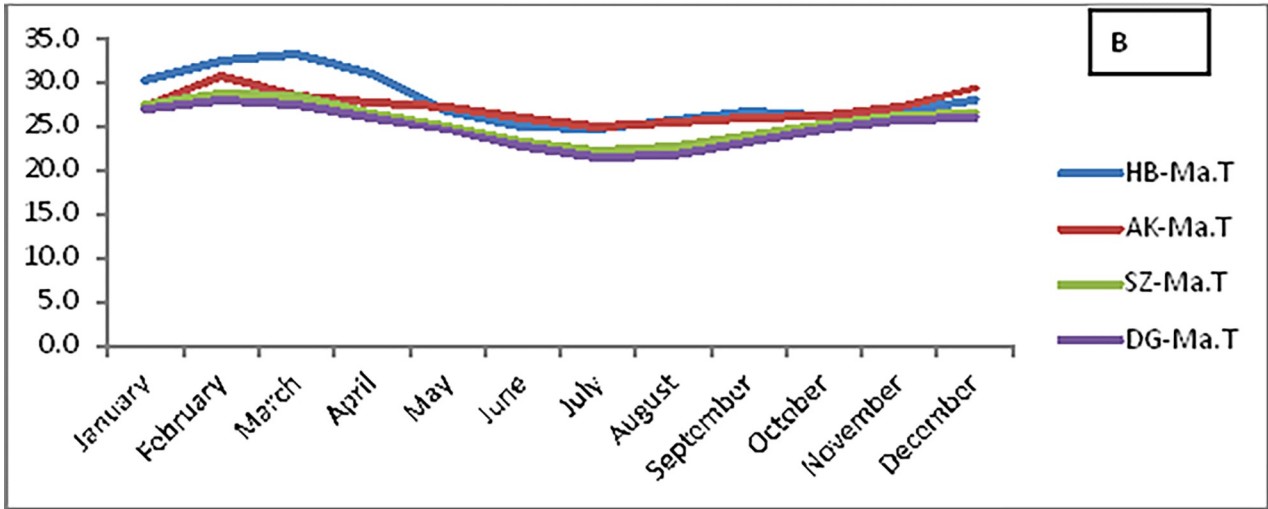

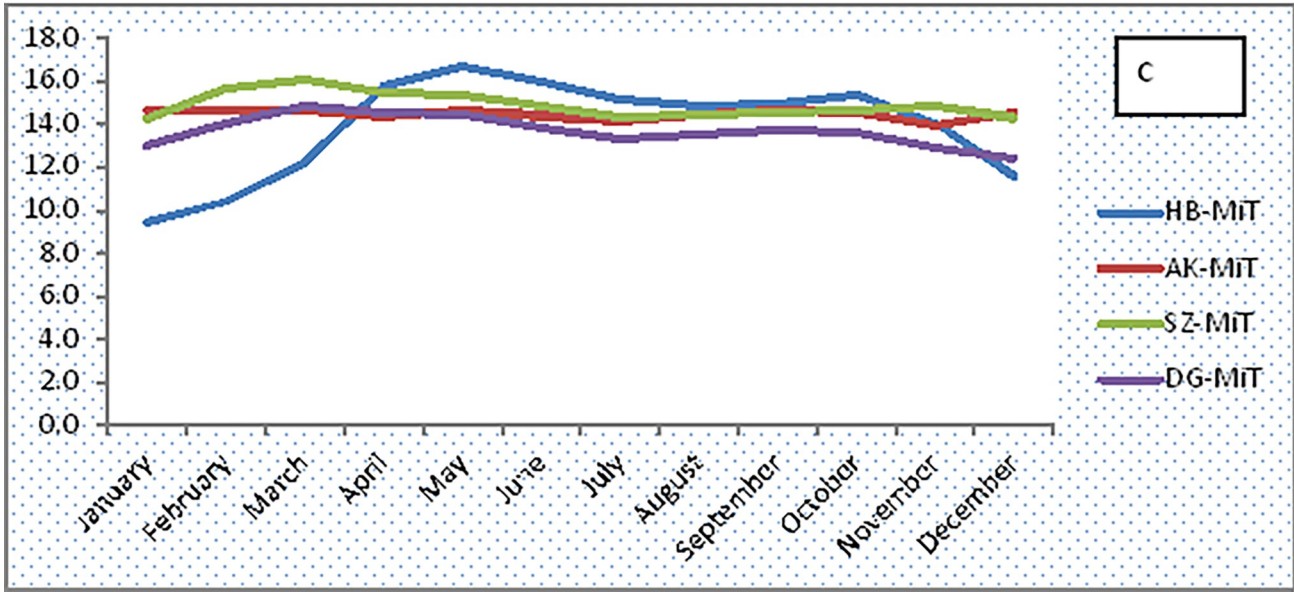

**Fig 8. Graphical presentation of 20 years average agro-climatic data of the four districts (HB = Humbo, AK = Boloso Sore, SZ = Sodo Zuria, and DG = Damot Gale) of Wolaita Zone.** A shows the rain fall pattern, B; maximum air temperature; C; minimum air temperature.

influenced by factors such as climate (Fig 8), altitude (Table 1), date of sowing, and soil type (Table 2). Genotypes with earlier heading and maturity may be preferred in regions with shorter growing seasons [44], while genotypes with late maturity may be suitable for longer growing seasons.

Mean performances of the 10 advanced bread wheat selections and Alidoro (G11) depicted that four of them G11, G9, G10, and G8 were outstanding in grain yield and its component traits such as thousand kernel weight, harvest index, aboveground dry biomass, number of kernels per spike and spike length. These findings are important for breeding programs aiming to improve crop productivity and adaptation in the study area. Other studies have also reported similar variations in those major traits [49].

Additionally, A study conducted by [50] reported that cultivar Alidoro performed best consistently in the northwest part of Amhara National Regional State, specifically in the Debark and Dabat districts in the North Gondar wheat growing belt. This suggests that G11 (Alidoro) could be a promising variety for farmers in those regions and other areas with similar agro-climatic conditions.

### 4.3. AMMI analysis

The ANOVA showed that the genotype, location, and GEI showed highly significant differences in grain yield (Table 8). This result was in harmony with the results of [51] on bread wheat performance evaluation with regards to genotype, environment, and GEI whereas contrary to this finding, the non-significant difference for genotype was reported by [46].

Still, the variance caused by locations makes up 82% of all variation and was over 9 times the variance caused by genotypes. The great variation in soil factors (soil properties such as; texture, pH, fertility, porosity, etc.) and weather patterns (amount and distribution of rainfall, minimum and maximum air temperatures) between locations as displayed in Table 2, and Fig 8, respectively, might have caused a higher proportion of the environmental variance. Yield variation was seen due to the environmental effect indicating that the environments were diverse and has become the major part of the variation in grain yield. Such kinds of views have been endorsed by several scholars including [52,53]. Consistent with this finding, [31] who evaluated 25 bread wheat genotypes in 9 rain-fed environments during the 2002–2003 growing season in Turkey reported that environment accounted for about 81% of all variation while genotypes accounted for about 7.3%.

However significant variations in GEI suggested that genotypes respond differently to various environments. Such findings indicate that genotypes, based on their grain yield capabilities, have distinct environmental adaptations. Several researchers reported that significant GEI showed that the reactions of the genotypes vary depending on environmental variables [54,55]. The current result is in agreement with the findings of [51,56] that showed only PCA1 was significant but PCA2 non-significant. Thus, the sum of squares due to GEI was mainly explained by the IPCA1 similar to the findings of [56].

### 4.4. Stability and adaptability analysis

The genotypes G5 and G8 were considered better yielding and stable genotypes in the current investigation because they were located closer to the line of IPCA1 than the others and on the upper right side of the grand mean level. A similar finding was reported by [57]. But genotypes

G9 and G10 performed best in the better locations and were included on the top high yielders next to the check genotype G11 based on the overall average yield data across locations, and confirmed superior in terms of grain yield performance and relatively stable according to the AMMI1 biplot analysis. Additionally, due to their proximity to the origin, genotypes G1, G3, and G4 were the most stable but low yielders (Fig 1).

In addition to this, the ASV and YSI values and rankings have been used to identify the most adapted and desirable genotypes across the wheat growing areas. The ASV is a calculated value used to rank and classify genotypes based on their yield stability [39]. ASV is derived using the first two principal component score values (IPCA1 and IPCA2) [47] and represents the distance of a genotype from the origin in a scatterplot; in addition the genotype with the lowest ASV is considered the most stable [21]. In the same manner, environments that scored low YSI values could be considered relatively stable. Therefore, we can conclude that Kokate and Damot Gale locations that exhibited low YSI values among the four locations could be considered as highly stable and favourable environments for wheat production (Table 9). This result is in conformity with an earlier study done by [58] who tried to determine the stability of 20 bread wheat genotypes.

Genotypes are thought to be environment-adaptive if their mean values are higher than the average (grand mean = 2.357t ha$^{-1}$) and their IPCA values are near zero, therefore, based on this criterion G5 (2.397t ha$^{-1}$, IPCA1 = 0.18) and G8 (2.569t ha$^{-1}$, IPCAI = 0.21) could be selected as the most adaptable genotypes (Table 9). However, genotypes with high average potential and high IPCA scores are probably have specific environment adaptations. As stated by [59], genotypes or environments with high IPCA1 scores, whether positive or negative, showed strong interactions, but genotypes with low IPCA1 values had smaller value for interactions. Similarly, [51] confirmed that the genotype IPCA scores from the AMMI analysis determine the stability of genotypes across locations.

## 4.5. GGE biplot analysis

In this study the polygon was connected by the five bread wheat advanced selections: G11, G2, G9, and G6 (Fig 3) that were the winner genotypes in the polygon, they are located far from the biplot origin, represent larger vectors from the origin and are more sensitive to GEI in each sector. These pairs of apex genotypes (G9 and G11) are the genotypes that perform best in each mega-environment, or poorest (G2 and G6) in some or all locations. Such views have been endorsed by several scholars [60], for instance, at Boloso Sore and Humbo, G11 was the most successful genotype, while G9 at Kokate and Damot Gale, hence, G11 won in the first mega environments, while G9 won in second mega environments. i.e., the indicated genotypes showed a specific adaptation to a nearby location(s). Since G2 and G6 positioned on the opposite side of all the four locations they are considered as undesired genotypes [61].

The pairs of locations; Boloso Sore and Humbo as the 1$^{st}$, and Damot Gale and Kokate as the 2$^{nd}$ pair, were identified as the two mega-environments to produce bread wheat genotypes, suggesting that equivalent genotypes perform better in a homogeneous location. The identified mega environments may therefore help manage GEIs and generalize outcomes to locations with similar agro-climatic conditions [60].

Similar to this study, [62] used a polygon approach to determine the performance of different genotypes in various environments. The genotypes at the edges of the polygon, known as Vertex genotypes, were the most sensitive [24]. The GGE biplot analysis, specifically the PC1 vs PC2 biplot, was used to identify the presence or absence of GEIs and to identify different mega-environments [63]. This analysis helps to assess the stability and suitability of genotypes for different environments [64]. In agreement to the current study, [65] who had evaluated 18 barley

genotypes in five different locations of Iran identified two locations; Gonbad and Moghan, as the most discriminating and representative environments using the GGE biplot analysis.

Generally, the study depicted presence of tangible differences among genotypes for most traits ($\geq$68.75%), suggesting possibility of improving those traits through selection and breeding of the genotypes investigated. Subsequently, differential response of the genotypes for grain yield has been revealed by the AMMI analysis, which further identified genotypes, G8, G9, G10, and G11 as relatively stable across locations and superior in grain yield performance (above the grand mean grain yield).

The three locations; Damot Gale, Boloso Sore, and Kokate showed above-average yields and are favorable for wheat crop cultivation. Some of the genotypes showed specific adaptation to a particular location, for instance, G10 at Damot Gale and G9 at Kokate, G2 at Humbo, while, the check genotype G11 at Boloso Sore, were the best genotypes for their corresponding locations due to specific stability. Genotypes G1, G3, and G4 were stable but low-yielders, suitable for less favorable conditions.

Locations Boloso Sore and Damot Gale were high contributors for GEIs, requiring further investigation, while Kokate was a low contributor for GEI and, hence, could be considered the most stable location for bread wheat cultivation. As for the amount of GEI, Humbo contributed less fulfilling the criterion for a stable location for wheat production, but the extremely low yield recorded by all genotypes indicated its inferiority or unsuitability for wheat production compared with the other locations, which could be attributed to its erratic rainfall distribution, relatively higher temperature during early growth and flowering stages of wheat accompanied by less amount of precipitation (Fig 8 and Table 2). These environmental discrepancies' could be the major causes of overall poor performance of the wheat genotypes including grain yield at Humbo site; such views has been reported by several researches as documented in [66]. Hence, a detailed study is needed to identify the exact potential of Humbo for wheat production; like using supplemental irrigation, varying planting dates, varietal selection for irrigated wheat production, heat tolerance, drought tolerance etc.

## 5. Conclusions

Generally, this study evaluated the performance and stability of new bread wheat genotypes compared to standard check Alidoro in Ethiopia's Wolaita zone. To this end, we identified high-yielding, stable, and adaptable cultivars, providing insights for selecting suitable wheat varieties to enhance domestic wheat breeding program and improve the productivity of wheat per unit area of land.

Since genotype G8 showed good performance and stability across all locations, we can recommend it for a potential seed source for wheat production across all the studied environments and similar agro-ecological zones. Damot Gale and Kokate are promising locations for wheat breeding programs due to their stability and high grain yield performance. However, further research is imperative to explore enhancement strategies and management practices, especially in specific locales like Humbo and Boloso Sore. This entails investigating irrigation methods, exploring biological interventions, or considering seasonal adjustments to optimize wheat cultivation in these areas. Such endeavors are crucial for refining wheat production techniques and ensuring sustainable agricultural practices in diverse geographical locations.

## Supporting information

**S1 Table. Summary of mean squares of 16 traits among 11 bread wheat genotypes tested at each location.**
(PDF)

**S2 Table. Mean values for different agronomic traits of the 11 genotypes at each environment in 2022.**
(PDF)

**S3 Table. Mean values for different agronomic traits of the 11 genotypes across environment in 2022.**
(PDF)

## Acknowledgments

We acknowledge the Institute of Biodiversity Conservation of Ethiopia for initial supply of the bread wheat landraces. We thank Wolaita Sodo University research and community service for hosting the research work and supplying the improved versions (S$_3$ selections) of the bread wheat seeds.

## Author Contributions

**Conceptualization:** Mesfin Kebede.

**Data curation:** Liyew Alemayehu, Mesfin Kebede.

**Formal analysis:** Liyew Alemayehu.

**Funding acquisition:** Mesfin Kebede.

**Investigation:** Liyew Alemayehu, Mesfin Kebede, Eyasu Wada.

**Methodology:** Mesfin Kebede.

**Project administration:** Mesfin Kebede.

**Resources:** Mesfin Kebede.

**Supervision:** Mesfin Kebede, Eyasu Wada.

**Validation:** Mesfin Kebede, Eyasu Wada.

**Visualization:** Mesfin Kebede.

**Writing – original draft:** Liyew Alemayehu.

**Writing – review & editing:** Mesfin Kebede, Eyasu Wada.

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
