## [Decision Letter · Decision Letter 0]

6 Sep 2024

PONE-D-24-31553AMMI Analysis of Elite Bread Wheat (Triticum aestivum L.) Selections for Genotype by Environment Interaction and Stability of Grain Yield in Southern EthiopiaPLOS ONE

Dear Dr. Gessese,

Thank you for submitting your manuscript to PLOS ONE. After careful consideration, we feel that it has merit but does not fully meet PLOS ONE’s publication criteria as it currently stands. Therefore, we invite you to submit a revised version of the manuscript that addresses the points raised during the review process.

We look forward to receiving your revised manuscript.

Kind regards,

Mehdi Rahimi, Ph.D.

Academic Editor

PLOS ONE

**Journal Requirements:**

4. We note that your Data Availability Statement is currently as follows: All relevant data are within the manuscript and its Supporting Information files

Reviewers' comments:

Reviewer's Responses to Questions

**Comments to the Author**

1. Is the manuscript technically sound, and do the data support the conclusions?

Reviewer #1: Yes

Reviewer #2: Partly

Reviewer #3: Yes

Reviewer #4: Partly

2. Has the statistical analysis been performed appropriately and rigorously? 

Reviewer #1: Yes

Reviewer #2: No

Reviewer #3: Yes

Reviewer #4: Yes

3. Have the authors made all data underlying the findings in their manuscript fully available?

Reviewer #1: Yes

Reviewer #2: No

Reviewer #3: Yes

Reviewer #4: No

4. Is the manuscript presented in an intelligible fashion and written in standard English?

Reviewer #1: Yes

Reviewer #2: No

Reviewer #3: Yes

Reviewer #4: Yes

5. Review Comments to the Author

**Reviewer #1:** The manuscript is well designed and written in a fluent language. Since the manuscript using AMMI and GGE biplot on bread wheat genotypes across four locations, it has an international audience. The statistical analysis is performed on the data and clearly discussed with the figures.

Therefore, the manuscript is suitable to publish on the PLOS ONE journal

**Reviewer #2:** • The authors didnt shows the aim of the research in the abstract section.

• For citation format, authors must re-read the author guidelines.

• Inconsistent in the units used (grain yield (t/ha), 2.67 t ha-1, and 2.51 ton ha-1), check in all sections for all units.

• It would be interesting to conduct a correlation analysis or path analysis between environmental factors (temperature, rainfall, altitudes, and soil conditions) with yield and yield attributes.

• It needs to be explained that AMMI, ASV, YSI, and GGE Biplot are used to identify stability. And why use these analyses? There are many other stability analyses that can be used, such as combined stability analysis (parametric and non-parametric). If combined between Visual Analysis (AMMI and GGE biplots) with parametric and non-parametric, it will be very interesting and more precise (authors can read and cite some of these articles: https://doi.org/10.1007/s10681-019-2386-5 and https://doi.org/10.34044/j.anres.2022.56.4.10)

• There is no information on soil conditions at each location. Because soil fertility is one of the factors that affect crop yields.

• The subtitle "Mean Performance of the 11 selections for the 16 traits evaluated across locations" is confusing.

• Inconsistent subtitles, such as: "Analysis of variance" uses sequential numbers, but other subtitles do not use sequential numbers. In addition, the subtitles are also less descriptive of what will be discussed and are too general.

• In table 5, the numbers after the decimal point are inconsistent.

• There are several typos, such as "spikelets", in several sentences it is written as "spiklets"

• Inconsistent paragraph writing.

• It is necessary to add data for one more planting season in each location so that the influence of the genotypes by environment interactions is more visible.

• The discussion is not strong. How did the authors determine the best genotype (Stable and high yield) from all the analyses used?

• The use of the abbreviation of Genotype by environment interactions is inconsistent. In some parts it is written as GxL, but in other parts it is written as GxE

• The authors did not reveal the limits of genotype stability using AMMI and ASV. In AMMI Biplot, the radius of the ellipse should be displayed, which is one indicator of genotype stability.

• For mega environment analysis, the data used is still lacking.

• The authors do not disclose how they drew conclusions from all the analyses used.

• The conclusion section is not clear/ not concrete.

**Reviewer #3:** My corrections are attached in the manuscript. The major corrections are as follow,

The location is contributing more to the variation, so it is recommended to provide soil analysis data and weather parameters month wise during cropping season (as the crop grown in rainfed condition month wise rainfall data provide more insights), if available.

The disease score were mentioned as percentage in text in result section, but in table-4 the unit given as scale. If the scale converted as percent disease incidence provide the methodology with formula in material and method.

Provide details of scale along with category of resistance/susceptibility.

As, mentioned in methodology the DMRT analysis was performed for traits having significance for genotype and GEI, but the LR trait showed non significant in combined ANOVA, but the mean comparison was performed as visualized from table-4. Provide the reason.

The Table-6 is a repetitive of Table-4 and 7, better it should be removed.

The table-7 is numbered as table-73, correct it.

The figures were numbered in Italic, change to Arabian numerals.

The Conclusion and recommendation section is more elaborative, which is repetitive of result and discussion. For better reading make it crisp by highlighting significant result and it's future implementation.

**Reviewer #4: **The topic and analysis is sound. However, I consider the number of years studied in this article to be insufficient to make conclusions. I suggest that the authors should gather at least one more year of data to present the article.

6. PLOS authors have the option to publish the peer review history of their article (what does this mean?). If published, this will include your full peer review and any attached files.

Reviewer #1: **Yes: **Ziya DUMLUPINAR

Reviewer #2: No

Reviewer #3: No

Reviewer #4: No

---

## [Author Response · Author response to Decision Letter 0]

19 Dec 2024

Authors Response to the Editor’s and Reviewers ’ Comments

AMMI analysis of elite bread wheat (Triticum aestivum L.) selections for genotype by environment interaction and stability of grain yield in Southern Ethiopia

Ref. MS. NO. PONE-D-24-31553

PLOS ONE

Editor’s comment: Dear Dr. Gessese, after careful consideration, we feel that it has merit but does not fully meet PLOS ONE’s publication criteria as it currently stands. Therefore, we invite you to submit a revised version of the manuscript that addresses the points raised during the review process. Please include the following items when submitting your revised manuscript: 1.A rebuttal letter that responds to each point rose by the academic editor and reviewer(s). You should upload this letter as a separate file labeled 'Response to Reviewers'. 2. A marked-up copy of your manuscript that highlights changes made to the original version. You should upload this as a separate file labeled 'Revised Manuscript with Track Changes'. 3. An unmarked version of your revised paper without tracked changes. You should upload this as a separate file labeled 'Manuscript'. The submitted documents should meet the journal’s requirements outlined in four points.

Dear respected Professor Mehidi Rahimi (Academic editor), we thank you very much for considering our work and to be critically evaluated and recommended it for revision and further consideration. We have tried our best to address all the suggestions, comments and made corrections to our mistakes following the guidelines provided by the editorial office and anonymous reviewers. In this rebuttal letter, we have provided point-by-point responses to all the comments for your reference and follow up of the review process. The constructive suggestions from the reviewers have significantly contributed to the refinement of the manuscript, and we, the authors, are thankful for their input. Furthermore, we have refined the language and expression within the manuscript to enhance its clarity and coherence. We prepared the new “manuscript” file following the guideline outlined by the editorial office so that it meets the journal’s standard. On behalf of all the authors, we request you to consider our manuscript for publication in your esteemed journal. Many thanks Sir for your valuable time and consideration.

Reviewer #1

The manuscript is well designed and written in a fluent language. Since the manuscript using AMMI and GGE biplot on bread wheat genotypes across four locations, it has an international audience. The statistical analysis is performed on the data and clearly discussed with the figures.

Therefore, the manuscript is suitable to publish on the PLOS ONE journal.

Author’s response: we are very much thankful for your review of our manuscript, consideration and further appreciation of the work. On behalf of all the authors, we request you to consider our manuscript for publication in this esteemed journal.

Reviewer #2

Comment #1; The authors didn’t show the aim of the research in the abstract section

Author’s response: dear reviewer, we authors are very grateful for your feedback on this part and we made the correction as indicated in the ‘revised manuscript with track changes’ file within line numbers (L#) 14-18.

Comment #2: For citation format, authors must re-read the author guidelines

Author’s response: Dear Reviewer, as per your remark we made all the necessary changes as shown in the new ‘manuscript’ file and supporting file.

Comment #3: Inconsistent in the units used (grain yield (t/ha), 2.67 t ha-1, and 2.51 ton ha-1), check in all sections for all units

Author’s response: Dear reviewer, thanking you again for your feedback, we made the corrections as indicated in lines 259, 367-369, 373, and elsewhere we found with the find and replace application of MS-Word program.

Comment #5: It needs to be explained that AMMI, ASV, YSI, and GGE Biplot are used to identify stability. And why use these analyses? There are many other stability analyses that can be used, such as combined stability analysis (parametric and non-parametric). If combined between Visual Analysis (AMMI and GGE biplots) with parametric and non-parametric, it will be very interesting and more precise (authors can read and cite some of these articles: https://doi.org/10.1007/s10681-019-2386-5, and https://doi.org/10.34044/j.anres.2022.56.4.10)

Author’s response: Dear reviewer, thanking you again for your constructive feedback, we tried to explain why we chose the AMMI, ASV, YSI, & GGE Biplot methods of analyses as shown in the introductory part of the manuscript line numbers; 94-120. We also read the article published by Vaezi et al. 2019 that was very interesting work where we found supporting evidences to our work and cited it in its appropriate place.

Comment #6: There is no information on soil conditions at each location. Because soil fertility is one of the factors that affect crop yields.

Author’s response: Dear reviewer, thanking you again for your insightful feedback, we collected the appropriate information/data, which appeared very important input to explain the environmental variation among the locations. We incorporated the soil properties data for each location in a tabulated form with caption “Table 2 Soil properties of the four districts of Wolaita zone that affect growth and development of agricultural crops”.

Comment #7: The subtitle "Mean Performance of the 11 selections for the 16 traits evaluated across locations" is confusing.

Author’s response: Dear reviewer, thanking you again for your insightful feedback, we tried to address the issue by modifying the title to make it clear as marked in the support information (revised manuscript with track changes’ file as indicated in L# 347-348.

Comment #8: Inconsistent subtitles, such as: "Analysis of variance" uses sequential numbers, but other subtitles do not use sequential numbers. In addition, the subtitles are also less descriptive of what will be discussed and are too general

Author’s response: Dear reviewer, thanking you for your important feedback, we corrected the mistakes and addressed the issue throughout the entire manuscript.

Comment #9: In table 5, the numbers after the decimal point are inconsistent.

Author’s response: Dear reviewer, thanking you for your important feedback, we corrected the mistakes & in Table 5 and other tables to maintain consistency.

Comment #10: There are several typos, such as "spikelets", in several sentences it is written as "spikelets".

Author’s response: Dear reviewer, thanking you for your important feedback, we corrected the mistakes throughout the whole document.

Comment #11: Inconsistent paragraph writing

Author’s response: Dear reviewer, thanking you for your important feedback, we corrected the mistakes following the journal’s guideline for paragraph writing

Comment #12: It is necessary to add data for one more planting season in each location so that the influence of the genotypes by environment interactions is more visible.

Author’s response: Dear reviewer, thanking you for your important feedback, we tried to address this issue by including additional data from previous preliminary yield trial data as shown in Table 3 in the ‘manuscript’ file. However; due to shortage of logistics and unforeseen circumstances we couldn’t further repeat the trials.

Comment #13: The discussion is not strong. How did the authors determine the best genotype (Stable and high yield) from all the analyses used?

Author’s response: Dear reviewer, thank you for your important feedback. We tried to identify the stable and high yielder genotypes as described in line numbers; L#548-554 & 570-573 and further modified the discussion with overall summarized information.

Comment#14: The use of the abbreviation of Genotype by environment interactions is inconsistent. In some parts it is written as GxL, but in other parts it is written as GxE

Author’s response: Dear reviewer, thank you for your important feedback. We made the correction except in the mean square table of the combined ANOVA.

Comment#15: The authors did not reveal the limits of genotype stability using AMMI and ASV. In AMMI Biplot, the radius of the ellipse should be displayed, which is one indicator of genotype stability

Author’s response: Dear reviewer, thank you for your feedback. We tried to address the issue by including Fig 6.

Comment#16: For mega environment analysis, the data used is still lacking.

Author’s response: Dear reviewer, thank you for your feedback. We understand your ideas regarding deficiencies in environments. Our response is shortage of logistics and other unforeseen circumstances.

Comment#17: The authors do not disclose how they drew conclusions from all the analyses used.

Author’s response: Dear reviewer, thank you for your comment. We tried to modify the conclusions part as shown in L#634-648 in the “revised manuscript with track changes” file.

Comment#18: The conclusion section is not clear/ not concrete.

Author’s response: Dear reviewer, thank you for your comment. We tried to modify the conclusion as stated above in comment#16.

Reviewer #3

Comment #1: The location is contributing more to the variation, so it is recommended to provide soil analysis data and weather parameters month wise during cropping season (as the crop grown in rain-fed condition month wise rainfall data provide more insights), if available.

Author’s response: Dear reviewer, thank you for your very constructive and critical comments that helped to elevate the quality of the manuscript. We have made all the corrections and tried to address the comments given by all team of reviewers given throughout the document. Regarding comment#1, we have included the soil analysis data and weather parameters month wise as per your recommendation.

Comment#2: The disease score were mentioned as percentage in text in result section, but in table-4 the unit given as scale. If the scale converted as percent disease incidences provide the methodology with formula in material and method. Provide details of scale along with category of resistance/susceptibility.

Author’s response: Dear reviewer, thank you for your very constructive and critical comments. We have included the information as table 2 and its explanation in line#256-261.

Comment #3: As, mentioned in methodology the DMRT analysis was performed for traits having significance for genotype and GEI, but the LR trait showed non-significant in combined ANOVA, but the mean comparison was performed as visualized from table-4. Provide the reason.

Author’s response: Dear reviewer, thank you for your very constructive and critical comments. We have corrected our mistake, it is a technical error.

Comment #4: The Table-6 is a repetitive of Table-4 and 7, better it should be removed.

The table-7 is numbered as table-73, correct it.

Author’s response: Dear reviewer, thank you again for your very constructive and critical comments. We have deleted table 6and corrected the mistake for wrongly numbered table.

Comment#5: The figures were numbered in Italic, change to Arabian numerals.

Author’s response: Dear reviewer, thank you again for your very constructive comments. We did as you suggested.

Comment #6: The Conclusion and recommendation section is more elaborative, which is repetitive of result and discussion. For better reading make it crisp by highlighting significant result and it's future implementation.

Author’s response: Dear reviewer, thank you again for your very constructive comments. We did as you suggested. Please kindly check it in L#634-648.

Reviewer #4: The topic and analysis is sound. However, I consider the number of years studied in this article to be insufficient to make conclusions. I suggest that the authors should gather at least one more year of data to present the article.

Author’s response: Dear reviewer, thank you very much for your very positive outlook and constructive comments. We tried to get the budget and other necessary logistics to repeat the field trials; however, it turned out to be negative response due to the shortage of budget and other unforeseen circumstances undergoing that create problems to do the trials.

---

## [Decision Letter · Decision Letter 1]

25 Dec 2024

PONE-D-24-31553R1AMMI Analysis of Elite Bread Wheat (Triticum aestivum L.) Selections for Genotype by Environment Interaction and Stability of Grain Yield in Southern EthiopiaPLOS ONE

Dear Dr. Gessese,

Thank you for submitting your manuscript to PLOS ONE. After careful consideration, we feel that it has merit but does not fully meet PLOS ONE’s publication criteria as it currently stands. Therefore, we invite you to submit a revised version of the manuscript that addresses the points raised during the review process.

We look forward to receiving your revised manuscript.

Kind regards,

Mehdi Rahimi, Ph.D.

Academic Editor

PLOS ONE

Journal Requirements:

Reviewers' comments:

Reviewer's Responses to Questions

**Comments to the Author**

1. If the authors have adequately addressed your comments raised in a previous round of review and you feel that this manuscript is now acceptable for publication, you may indicate that here to bypass the “Comments to the Author” section, enter your conflict of interest statement in the “Confidential to Editor” section, and submit your "Accept" recommendation.

Reviewer #3: All comments have been addressed

2. Is the manuscript technically sound, and do the data support the conclusions?

Reviewer #3: Yes

3. Has the statistical analysis been performed appropriately and rigorously? 

Reviewer #3: Yes

4. Have the authors made all data underlying the findings in their manuscript fully available?

Reviewer #3: Yes

5. Is the manuscript presented in an intelligible fashion and written in standard English?

Reviewer #3: Yes

6. Review Comments to the Author

Reviewer #3: The authors have carefully addressed all the recommendations as suggested. However, a few recommendations need to be carried out before the manuscript (MS) is published.

The description of the testing environments given under Section 2.1 is too descriptive and deviates from the scope of the journal. Therefore, I suggest that the authors add a single paragraph explaining the key information about the sites, as the other climatic and soil parameters are provided in the table.

In Section 2.3, the section number is missing for the data collection subheading. Additionally, the traits listed under this subheading are in italics; please change them to regular font.

In the results under Sections 3.1 and 3.2, the traits can be mentioned in short form, with abbreviations explained in the materials section.

7. PLOS authors have the option to publish the peer review history of their article (what does this mean?). If published, this will include your full peer review and any attached files.

Reviewer #3: No

---

## [Author Response · Author response to Decision Letter 1]

31 Dec 2024

Authors Response to the Editor’s and Reviewers ’ Comments

AMMI analysis of elite bread wheat (Triticum aestivum L.) selections for genotype by environment interaction and stability of grain yield in Southern Ethiopia

Ref. MS. NO. PONE-D-24-31553

PLOS ONE

Editor’s comment: Dear Dr. Gessese,

Thank you for submitting your manuscript to PLOS ONE. After careful consideration, we feel that it has merit but does not fully meet PLOS ONE’s publication criteria as it currently stands. Therefore, we invite you to submit a revised version of the manuscript that addresses the points raised during the review process. Please include the following items when submitting your revised manuscript:

Dear respected Professor Mehidi Rahimi (Academic editor), we thank you very much again for considering our work and recommended it for minor revision and further consideration. We have addressed all the suggestions, comments and made corrections to the mistakes requested by reviewer#3. In this rebuttal letter, we have provided point-by-point responses to all the comments provided by reviewer#3 for your reference and follow up of the review process. We prepared the new “manuscript” file following the guideline outlined by the editorial office so that it meets the journal’s standard. On behalf of all the authors, we request you to consider our manuscript for publication in your esteemed journal. Many thanks Sir for your valuable time and further consideration.

Reviewer #3: 

Comment#1: The authors have carefully addressed all the recommendations as suggested. However, a few recommendations need to be carried out before the manuscript (MS) is published. The description of the testing environments given under Section 2.1 is too descriptive and deviates from the scope of the journal. Therefore, I suggest that the authors add a single paragraph explaining the key information about the sites, as the other climatic and soil parameters are provided in the table.

Author’s response: dear reviewer, we authors are very grateful for your unreserved effort to improve this manuscript and deliver your feedback. Based on your suggestion we made the correction as shown in the file ‘Revised manuscript with track changes’ within line numbers (L#) 134-171, 

Comment#2: In Section 2.3, the section number is missing for the data collection subheading. Additionally, the traits listed under this subheading are in italics; please change them to regular font.

Author’s response: dear reviewer, thanks again for your comment we made the correction as shown in the file ‘Revised manuscript with track changes’ within line numbers (L#) 259-271.

Comment#3: In the results under Sections 3.1 and 3.2, the traits can be mentioned in short form, with abbreviations explained in the materials section.

Author’s response: dear reviewer, thanks again for your comment we made the correction as shown in the file ‘Revised manuscript with track changes’ within line numbers (L#) 351-354, 362-364, 371-372, 379, 383-385, 387-388, 394.

---

## [Decision Letter · Decision Letter 2]

19 Jan 2025

AMMI Analysis of Elite Bread Wheat (Triticum aestivum L.) Selections for Genotype by Environment Interaction and Stability of Grain Yield in Southern Ethiopia

PONE-D-24-31553R2

Dear Dr. Gessese,

We’re pleased to inform you that your manuscript has been judged scientifically suitable for publication and will be formally accepted for publication once it meets all outstanding technical requirements.

Kind regards,

Mehdi Rahimi, Ph.D.

Academic Editor

PLOS ONE

Additional Editor Comments (optional):

Reviewers' comments:

Reviewer's Responses to Questions

**Comments to the Author**

1. If the authors have adequately addressed your comments raised in a previous round of review and you feel that this manuscript is now acceptable for publication, you may indicate that here to bypass the “Comments to the Author” section, enter your conflict of interest statement in the “Confidential to Editor” section, and submit your "Accept" recommendation.

Reviewer #3: All comments have been addressed

2. Is the manuscript technically sound, and do the data support the conclusions?

Reviewer #3: Partly

3. Has the statistical analysis been performed appropriately and rigorously? 

Reviewer #3: Yes

4. Have the authors made all data underlying the findings in their manuscript fully available?

Reviewer #3: Yes

5. Is the manuscript presented in an intelligible fashion and written in standard English?

Reviewer #3: Yes

6. Review Comments to the Author

Reviewer #3: The authors have included all the recommended comments, hence the article can be accepted for publication

7. PLOS authors have the option to publish the peer review history of their article (what does this mean?). If published, this will include your full peer review and any attached files.

Reviewer #3: No

---

## [Editor Report · Acceptance letter]

22 Jan 2025

PONE-D-24-31553R2 

PLOS ONE

Dear Dr. Gessese, 

I'm pleased to inform you that your manuscript has been deemed suitable for publication in PLOS ONE. Congratulations! Your manuscript is now being handed over to our production team.

Kind regards, 

on behalf of

Associate Prof. Mehdi Rahimi 

Academic Editor

PLOS ONE